# A primate nigrostriatal atlas of neuronal vulnerability and resilience in a model of Parkinson's disease

Lei Tang[1,6], Nana Xu[1,6], Mengyao Huang[1,6], Wei Yi[1,6], Xuan Sang[1,6], Mingting Shao[1], Ye Li[1], Zhao-zhe Hao[1], Ruifeng Liu[1], Yuhui Shen[1], Feng Yue[2,3], Xialin Liu ◉[1] ✉, Chuan Xu ◉[4] ✉ & Sheng Liu ◉[1,5] ✉

The degenerative process in Parkinson's disease (PD) causes a progressive loss of dopaminergic neurons (DaNs) in the nigrostriatal system. Resolving the differences in neuronal susceptibility warrants an amenable PD model that, in comparison to post-mortem human specimens, controls for environmental and genetic differences in PD pathogenesis. Here we generated high-quality profiles for 250,173 cells from the substantia nigra (SN) and putamen (PT) of 1-methyl-4-phenyl-1,2,3,6-tetrahydropyridine (MPTP)-induced parkinsonian macaques and matched controls. Our primate model of parkinsonism reca-pitulates important pathologic features in nature PD and provides an unbiased view of the axis of neuronal vulnerability and resistance. We identified seven molecularly defined subtypes of nigral DaNs which manifested a gradient of vulnerability and were confirmed by fluorescence-activated nuclei sorting. Neuronal resilience was associated with a *FOXP2*-centered regulatory pathway shared between PD-resistant DaNs and glutamatergic excitatory neurons, as well as between humans and nonhuman primates. We also discovered acti-vation of immune response common to glial cells of SN and PT, indicating concurrently activated pathways in the nigrostriatal system. Our study pro-vides a unique resource to understand the mechanistic connections between neuronal susceptibility and PD pathophysiology, and to facilitate future bio-marker discovery and targeted cell therapy.

PD, a common neurodegenerative disorder affecting 2–3% of the human population older than 65, is predominantly characterized by the loss of DaNs in the SN and by the deficit of dopamine primarily in the PT, an axonal projection area of DaNs[1–4]. Historically, mouse models of PD provided critical clues for understanding the underlying pathogenesis. Nevertheless, to what extent the rodent models recapitulate the disease complexity of PD in humans remains questionable[5].

Use of post-mortem human brains, in particular the SN region, has begun to unveil the cellular and molecular underpinnings of PD by means of immunohistology, genome wide association studies, and more recently, large-scale transcriptomic profiling of single nuclei

[1]State Key Laboratory of Ophthalmology, Zhongshan Ophthalmic Center, Sun Yat-sen University, Guangdong Provincial Key Laboratory of Ophthalmology and Visual Science, Guangzhou, China. [2]State Key Laboratory of Digital Medical Engineering, School of Biomedical Engineering, Hainan University, Haikou 570228, China. [3]Key Laboratory of Biomedical Engineering of Hainan Province, One Health Institute, Hainan University, Haikou 570228, China. [4]Wellcome Sanger Institute, Wellcome Genome Campus, Hinxton, Cambridge, UK. [5]Guangdong Province Key Laboratory of Brain Function and Disease, Guangzhou, China. [6]These authors contributed equally: Lei Tang, Nana Xu, Mengyao Huang, Wei Yi, Xuan Sang. ✉e-mail: liuxl28@mail.sysu.edu.cn; cx1@sanger.ac.uk; liush87@mail.sysu.edu.cn

(snRNA-seq)[6–9]. However, post-mortem brains usually reflect the end-stage of PD when DaNs tend to be undersampled in the human specimens[9–11]. Moreover, the etiology of PD is multifactorial including influences from aging, genetic, and environmental factors[1,12–15], thereby confounding the interpretability of case-control comparisons even with matched age and sex.

As a promising surrogate, the non-human primate model of parkinsonism induced by MPTP reproduces most of the clinical and pathological manifestations of PD such as depletion of DaNs[16–18]. With a strict control of environmental and other variations, molecular and cellular differences between parkinsonian and control individuals can be investigated in an unbiased manner. Another advantage of the primate model of parkinsonism is that it can avoid the postmortem effects in comparison with human samples. This is particularly important for studying the heterogeneity and vulnerability of DaNs, as DaNs harbor a selective susceptibility to neuronal demise across different subtypes regardless of their general loss in PD[10]. Furthermore, how cell types respond to PD in the PT, the region to which DaNs primarily project, is still unclear.

Here we leveraged our primate model, in combination with the single-cell/nucleus profiling technology, to survey the transcriptomic changes associated with PD pathology, providing a nonhuman primate atlas in the nigrostriatal system of SN and PT. We uncovered a transcriptional gradient of DaNs which showed selective vulnerability or resilience to neuronal loss in PD. In contrast to the SOX6+ PD-vulnerable DaNs, PD-resistant subtypes of DaNs uniquely expressed genes such as *SORCS3* and *FOXP2* which were also overrepresented in the glutamatergic excitatory neurons with higher resistance to neuronal loss. Cross-species comparison further showed that the regulons they formed can robustly define the degree of DaNs resilience in both humans and macaques.

## Results

### Molecular taxonomy of nigrostriatal cells in health and parkinsonism

We sampled the SN and PT sections from six adult *Macaca fascicularis* for both scRNA-seq and snRNA-seq using the 10x Genomics Chromium platform (Fig. 1a). Among them, two macaques were treated with MPTP and manifested stable PD-like symptoms including bradykinesia, rigidity and tremor, with the remaining four used as match controls (Supplementary Fig. 1a and Supplementary Data 1). Immunohistological examination of tyrosine hydroxylase (TH) confirmed the loss of DaNs in the macaque model of parkinsonism ($p < 0.0001$) (Fig. 1a and Supplementary Fig. 1b). Similarly, immunostainings of glial markers (IBA1, GFAP and OLIGO2) revealed a significant microgliosis ($p < 0.0001$) (Supplementary Fig. 2a), astrogliosis ($p < 0.0001$) (Supplementary Fig. 2b), and oligodendrogliosis ($p < 0.05$) (Supplementary Fig. 2c) in the MPTP-treated subjects. After a quality control procedure, a total of 250,173 cells (SN, 159,316; PT, 90,857) were obtained from the 31 samples generated (Fig. 1b), which were well-represented across regions and disease conditions with an average of 6572 transcripts and 2438 genes expressed per cell (Supplementary Fig. 3a, b).

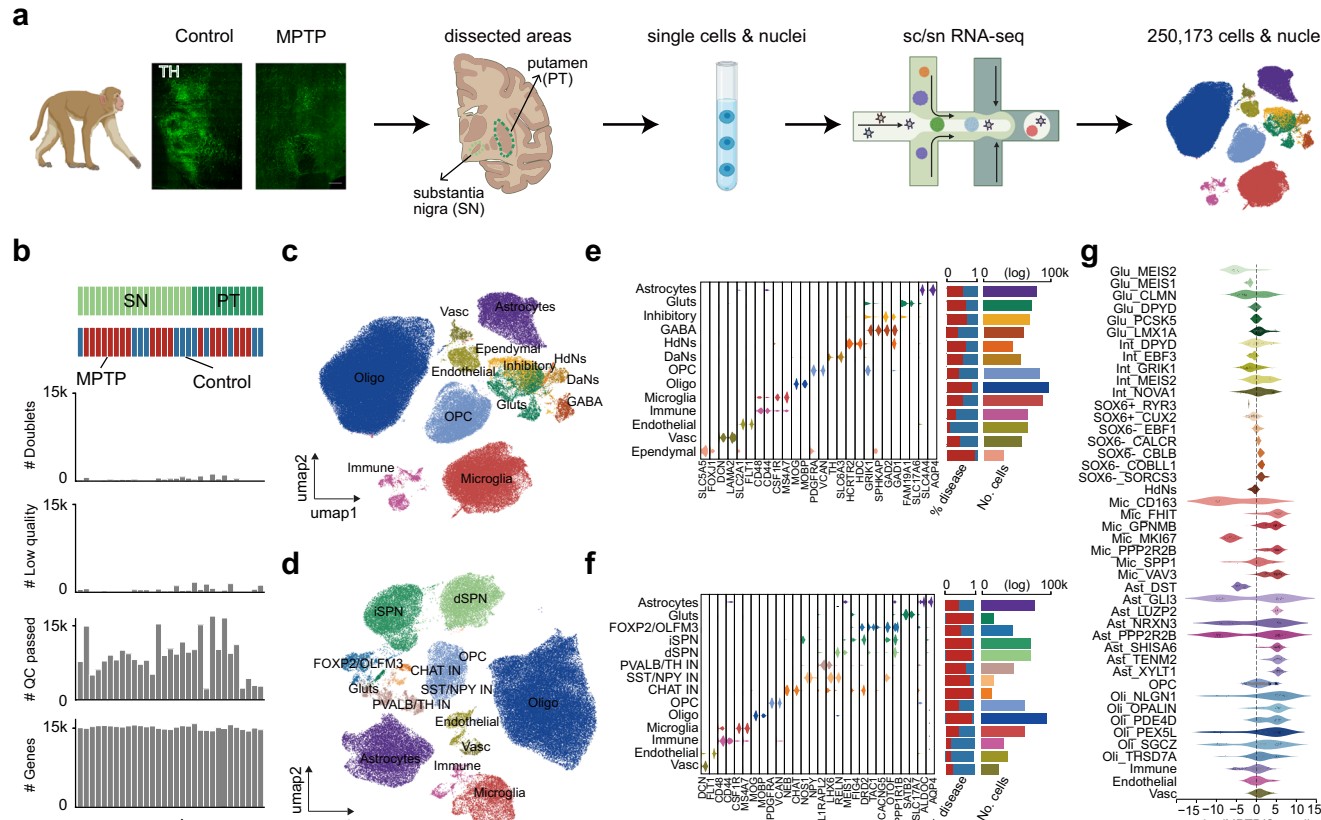

**Fig. 1 | A primate nigrostriatal atlas for studying Parkinson's disease.**
**a** Schematic showing single-cell and single-nucleus profiling of macaque substantia nigra (SN) and putamen (PT) under health and parkinsonism. **b** Upper: distribution of the 31 brain samples across regions and disease conditions. Lower: bar plots depicting, from top to bottom, the numbers of doublets, low- and high-quality cells, and detected genes across samples. **c** UMAP visualization of SN cells colored by cell types. **d** As in (**c**), but for PT cells. **e** Left: violin plot displaying expression of discriminative marker genes across major cell types. Right: two bar plots showing the relative contributions of disease conditions (left) and cell abundance (right) for each cell type. **f** As in (**e**), but for cells in the PT region. **g** Violin plot showing cell abundance changes between parkinsonian and control subjects, with the dots representing neighborhoods grouped by cell types. Source data are provided as a Source Data file.

We next mapped the molecular taxonomy of these cells and identified 13 and 14 transcriptionally distinct cell types in the SN and PT, respectively (Fig. 1c, d and Supplementary Fig. 3c, d). The transcriptomic structures of cell clusters were biologically driven and exhibited a minimal impact from the technical confounders after batch correction (Supplementary Fig. 3e). Eight cell types were shared between the SN and PT, including glutamatergic excitatory neurons (Gluts) (*SLC17A6*[+]) and inhibitory neurons (*GAD1*[+]), oligodendrocytes (Oligo) (*MOBP*[+]), oligodendrocyte precursor cells (OPC) (*VCAN*[+]), astrocytes (*AQP4*[+]), microglia (*CSF1R*[+]) and immune cells (*LSP1*[+], *CD3G*[+]), endothelial cells (*FLT1*[+]) and other types of vascular cells (Fig. 1e, f and Supplementary Fig. 4). In the SN, we detected the presence of *SLC6A3*- and *TH*-expressing DaNs as expected, as well as GABA (*GAD1*[+], *GRIK1*[+]) neurons[11]. In addition, we located a separated neuronal cluster (HdNs) in the SN characterized by overexpression of *HDC*, a gene catalyzing the biosynthesis of histamine (Fig. 1e). A population of ependymal cells (*FOXJ1*[+]) was also found in the SN. In the PT, we recovered a great diversity of cell types, consistent with those reported by a previous study on the macaque striatum[19] (Fig. 1f). Specifically, we detected two types of spiny projection neurons (SPNs) including the classical direct (dSPNs, *DRD1*[+]) and indirect (iSPNs, *DRD2*[+]) neurons (Fig. 1f). We also detected four types of inhibitory neurons including *SST*[+]*NPY*[+], *PVALB*[+], *CHAT*[+], and *FOXP2*[+]*OTOF*[+] inhibitory neurons, in keeping with previously detected inhibitory neuronal populations in mouse and human PT[20,21]. Of note, the proportions of cell types were consistent with those from previous studies[9], demonstrating the robustness of our strategy in cell sampling and inclusion under pathological statuses (Supplementary Fig. 3d).

We further assessed the compositions of neuronal cell types in our data by only focusing on the nuclei, as nuclei are less prone to damage than cells during isolation and thus maintain a better representation of cell types. This analysis showed that in comparison to glial cells which constituted most of the nuclei, neurons accounted for 7.9% and 29.8% in the SN and PT, respectively (Fig. 1e, f). DaNs made up 0.56% of all nuclei in the SN, representing 0.52% ± 0.46% of cells per individual, a number higher than other snRNA-seq studies with an unbiased dissection protocol[9,11]. As expected, in the SN, MPTP-treated monkeys had fewer DaNs, HdNs and Gluts than those from the controls (Supplementary Fig. 5). We also observed an aberrant increase in the numbers of microglia, astrocytes and oligodendrocytes in parkinsonian monkeys (Supplementary Fig. 5), consistent with the reported microgliosis and astrogliosis in PD pathogenesis[11,22,23].

To probe into the finer details of cell proportional changes caused by MPTP-induced parkinsonism, we sub-clustered each broad cell category into molecularly defined subtypes in the SN, resulting in a total of 44 cell subtypes (Supplementary Fig. 6). On the basis of this, we performed differential abundance analysis[24] between parkinsonian individuals and controls, and discerned several subtypes enriched or depleted by the MPTP treatment (Fig. 1g). For example, we noticed an increase in the *GPNMB*-expressing microglial subset that was previously reported to be PD-associated[11,25]. The favorable cell quality and cell type coverage of our data allowed us to further pinpoint its pathogenic origin. Specifically, the abundance of a proliferating (*MKI67*[+]) microglial subtype was found to be significantly lower after MPTP treatment, indicating a putative trajectory toward functional PD-associated microglia (e.g., the *GPNMB*[+] or *VAV3*[+] microglia) by quickly consuming this progenitor population (Supplementary Fig. 7). Since neurons (in particular, DaNs) were expected to demonstrate the strongest differential abundance between MPTP-treated and control subjects, we focused on the neuronal cells in the section below.

## Differential vulnerability and resilience of DaNs

DaNs are the most affected cell type in PD. To investigate the transcriptional heterogeneity of DaNs, we in silico extracted the 730 DaNs in the SN, reclustered them, and identified seven transcriptomic

subpopulations (Fig. 2a). These subtypes likely reflected a genuine division of DaNs in the SN of healthy and parkinsonian macaques, given that no PD-specific sub-clusters were found and no significant difference existed between healthy and MPTP-treated subjects in the sample of origin, cell number, and various batch effects (Supplementary Fig. 8a–c).

We found a clear demarcation between the *SOX6*[+] and *SOX6*[−] groups of DaNs (Fig. 2b), aligning in part with the *SOX6-CALB1* axis reported in humans (*CALB1* was however not enriched in the *SOX6*[−] DaNs in our macaque data, as is the case for another macaque study[10]). Cells from the two *SOX6*[+] subtypes were contributed predominantly by healthy subjects, as affirmed statistically by a differential abundance bias toward macaque controls (Fig. 2b). Therefore, the two *SOX6*[+] subtypes represented the vulnerable DaNs in the macaque SN. In addition, they highly expressed *KCNJ6*, a marker gene relating to differential vulnerability in PD[26] (Fig. 2b and Supplementary Fig. 8d), and *GRM5*, inhibition of which has been shown to have protection effects on DaNs[27], as well as other vulnerability markers including *ALDH1A1* and *SOX6*[28] (Supplementary Fig. 8d). Notably, among published PD and AD risk genes (Supplementary Data 2), we found four PD risk genes (*SH3GL2*, *LRRK2*, *SATB1*, *BST1*)[29,30], and an Alzheimer's disease risk gene (*CR1*)[31], which were highly expressed in these vulnerable DaNs subclusters (Supplementary Fig. 8d).

By contrast, the five *SOX6*[−] DaNs subtypes displayed a proportional increase in numbers in MPTP-treated subjects, characteristic of PD-resistant DaNs with varying degrees of resilience (Fig. 2b). Differential expression analysis revealed the widespread expression of *FOXP2* across the five *SOX6*[−] subtypes when compared to the *SOX6*[+] populations (Fig. 2b). Interestingly, an independent exploration of their underlying regulatory networks highlighted the FOXP2 regulon as well (Fig. 2c, d and Supplementary Data 3), suggesting a highly convergent regulatory unit composed of FOXP2 and the genes it regulates such as *TMEFF2* and *RELN* (Fig. 2e). *RELN* encodes an extracellular glycoprotein that is involved in dopamine neuron migration, positioning, and dendritic spine growth[32]. A recent study also indicated its protective role in dopaminergic neurodegeneration[33]. *TMEFF2*, which mitigates neuronal death both in vitro and in vivo in several brain disorders[34,35], was shown to promote survival of mesencephalic dopaminergic neurons[36]. We also visualized the topology of FOXP2 regulation, and prioritized target genes according to their importance scores (Fig. 2c). Among these FOXP2-regulating populations, a *SORCS3*-expressing subtype, SOX6[−]_SORCS3, demonstrated the strongest resilience to neurodegeneration induced by MPTP (Fig. 2b). This subtype was shown to bear a strong regulatory impact from the NR3C1 regulon, coinciding with its top hit as *SORCS3* (Fig. 2c, d).

To further verify the identity and confirm the robustness of DaNs identified in our study, we leveraged fluorescence-activated nuclei sorting (FANS) as described in a previous study[10] to enrich DaNs by selecting NR4A2[+]DAPI[+] cells in the SN (Supplementary Fig. 9a). Through this, we generated transcriptomic profiles for 31,793 cells corresponding to 11 broad cell types from one MPTP-treated macaque and one age-matched control (Supplementary Fig. 9b–d). Among them, 758 DaNs were identified (Supplementary Fig. 9e). Thus, with FANS we enriched a much higher proportion of DaNs (2.19% ± 0.58%) to complement those obtained earlier (0.52% ± 0.46%) (Supplementary Fig. 9f).

All DaNs subtypes detected in the previous snRNA-seq dataset were recapitulated by the 758 DaNs derived from our sorting strategy (Supplementary Fig. 10a), as also supported by combinatorial expression of multiple marker genes (Supplementary Fig. 10b). For example, the *SOX6-FOXP2* expression gradient separating the vulnerable and resilient DaNs subclusters was captured by these DaNs (Supplementary Fig. 10c). We also characterized the expression of a number of defining markers including genes regulated by the FOXP2 and NR3C1 regulons, which was again reminiscent of the

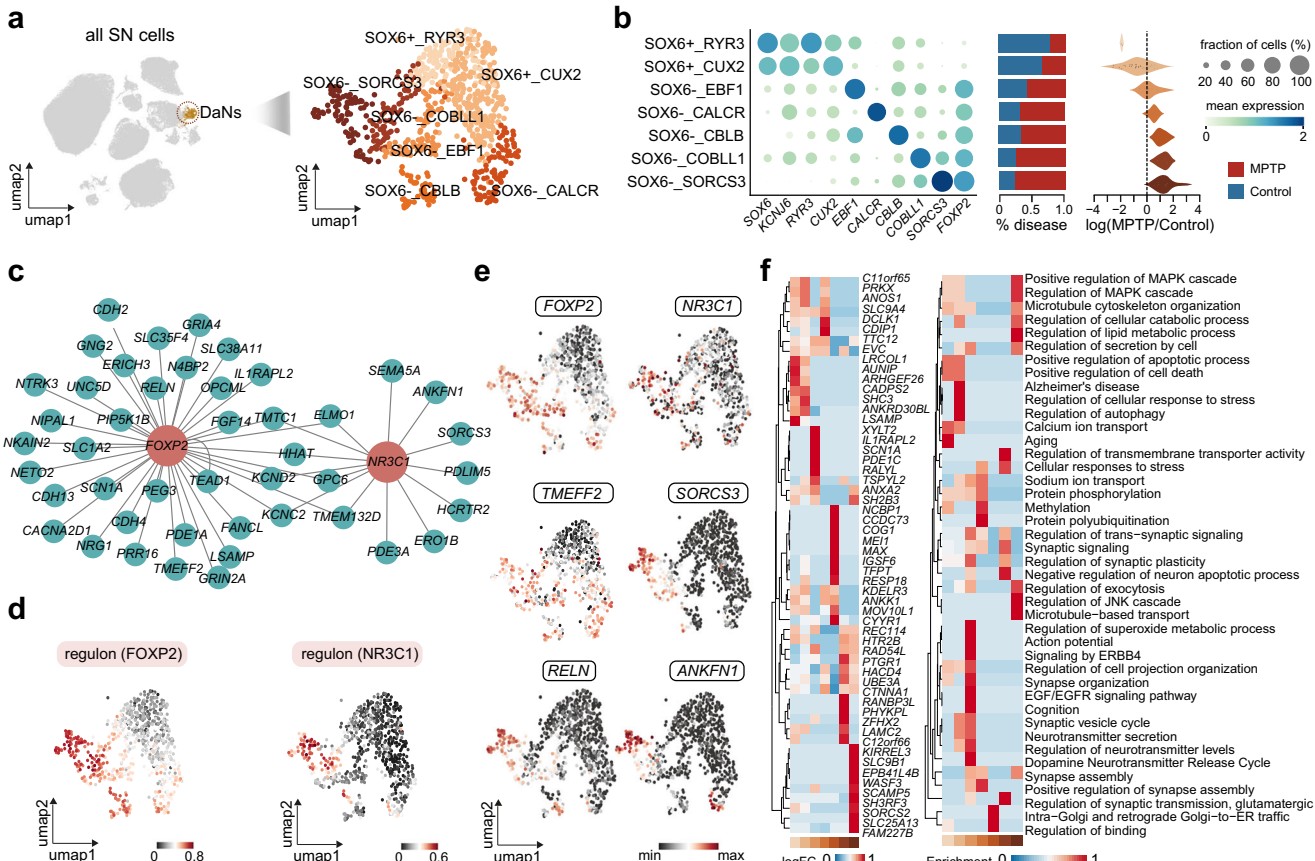

**Fig. 2 | Dopaminergic neurons in the substantia nigra display a transcriptomic heterogeneity and differential susceptibility to Parkinson's disease. a** Left: UMAP visualization of all cells in the substantia nigra. Inset shows the dopaminergic neurons (DaNs). Right: UMAP visualization of DaNs colored by the seven subtypes identified, with darker color indicating higher neuronal resilience to MPTP-induced parkinsonism. **b** Left: dot plot showing expression of selected marker genes across DaNs subtypes. Color of the dot represents normalized gene expression, and size represents the percentage of cells expressing a given gene. Middle: stacked bar plot exhibiting the proportions of cells in normal (blue) and diseased (red) conditions. Right: violin plot showing cell abundance changes between MPTP-treated and control subjects, with the dots representing neighborhoods grouped by cell subtypes. **c** Topological structure of the FOXP2 (left) and NR3C1 (right) regulons. **d** UMAP visualizations of DaNs colored by activity scores of FOXP2 (left) and NR3C1 (right) regulons. **e** UMAP visualizations of DaNs colored by expression of selected genes within the FOXP2 (left) and NR3C1 (right) regulons. **f** Left: heatmap depicting expression of differentially expressed genes across the seven DaNs subtypes. Right: heatmap depicting enrichment scores of differentially enriched gene ontology terms across the seven DaNs subtypes. Source data are provided as a Source Data file.

transcriptomic structure of cells from our unsupervised dissection protocol (Supplementary Fig. 10d). As expected, comparison of the abundance of DaNs subtypes between control and parkinsonian macaques revealed a greater proportion of *FOXP2* + DaNs and a smaller percentage of *SOX6* + DaNs in parkinsonian macaques (Supplementary Fig. 10b, c, e). This was further consistent with the PD-resistant and -susceptible roles of *FOXP2* and *SOX6* we observed earlier, respectively.

To map the identified molecular gradient of neuronal resilience within a spatial context, we performed immunoprofiling of these DaNs subtypes in normal brain tissue, and further coupled the resulting in vivo distribution with previously published Slide-seq dataset in macaque SN[10]. KCNJ6, the marker we identified to be enriched in the vulnerable *SOX6*+ DaNs, was co-stained with TH in a great density, indicating widespread existence of vulnerable neurons under the normal condition (Supplementary Fig. 11a). These DaNs were shown to populate the ventral tier of SN in vivo. Confirming this, re-analysis of the macaque Slide-seq data revealed their spatial localization in the ventral tier of SN (Supplementary Fig. 11b, c). We also immunostained the resilience marker FOXP2, and found that these resilient DaNs were distributed mainly in the dorsal and midline tier (Supplementary Fig. 11d), consistent with the distribution of *CALB1*+ resilient DaNs identified in an independent study[37]. Lastly, we focused on the most resilient DaNs subtype SOX6⁻_SORCS3, and revealed its enrichment in

the dorsal tier suggested by both immunostainings of SORCS3 (Supplementary Fig. 11e) and cell type deconvolution of spatial data (Supplementary Fig. 11b, f).

We next set out to uncover the molecular alterations driving the transition of DaNs from homeostatic to pathologic states, and identified a total of 818 genes that were differentially expressed between MPTP-treated and control macaques in at least one DaNs subtype (Supplementary Data 4). This analysis revealed both shared and cell-type-specific deviations in gene expression (Fig. 2f). The vulnerable DaNs subtypes upregulated an extensively overlapped set of genes, indicating a similar response induced by PD pathology. Gene ontology (GO) enrichment analysis showed that these genes converged on pathways relating to neuron apoptosis progress and calcium ion transport (Fig. 2f). For example, the two subtypes of vulnerable DaNs overexpressed *CADPS2* in the PD state, which was reported to drive the identity of a specific degenerating subtype in idiopathic PD[11] (Fig. 2f). To a lesser extent, we observed concurrently perturbed genes in the resilient DaNs, which were functionally enriched in "regulation of secretion by cell" and "regulation of exocytosis" (Fig. 2f). Subtype-specific expression changes were also observed, such as enhanced expression of *TGFB1* after the MPTP treatment for the most resilient subtype SOX6⁻_SORCS3. *TGFB1* was reported to suppress the neuroinflammatory responses and delay the neurodegeneration in PD[38].

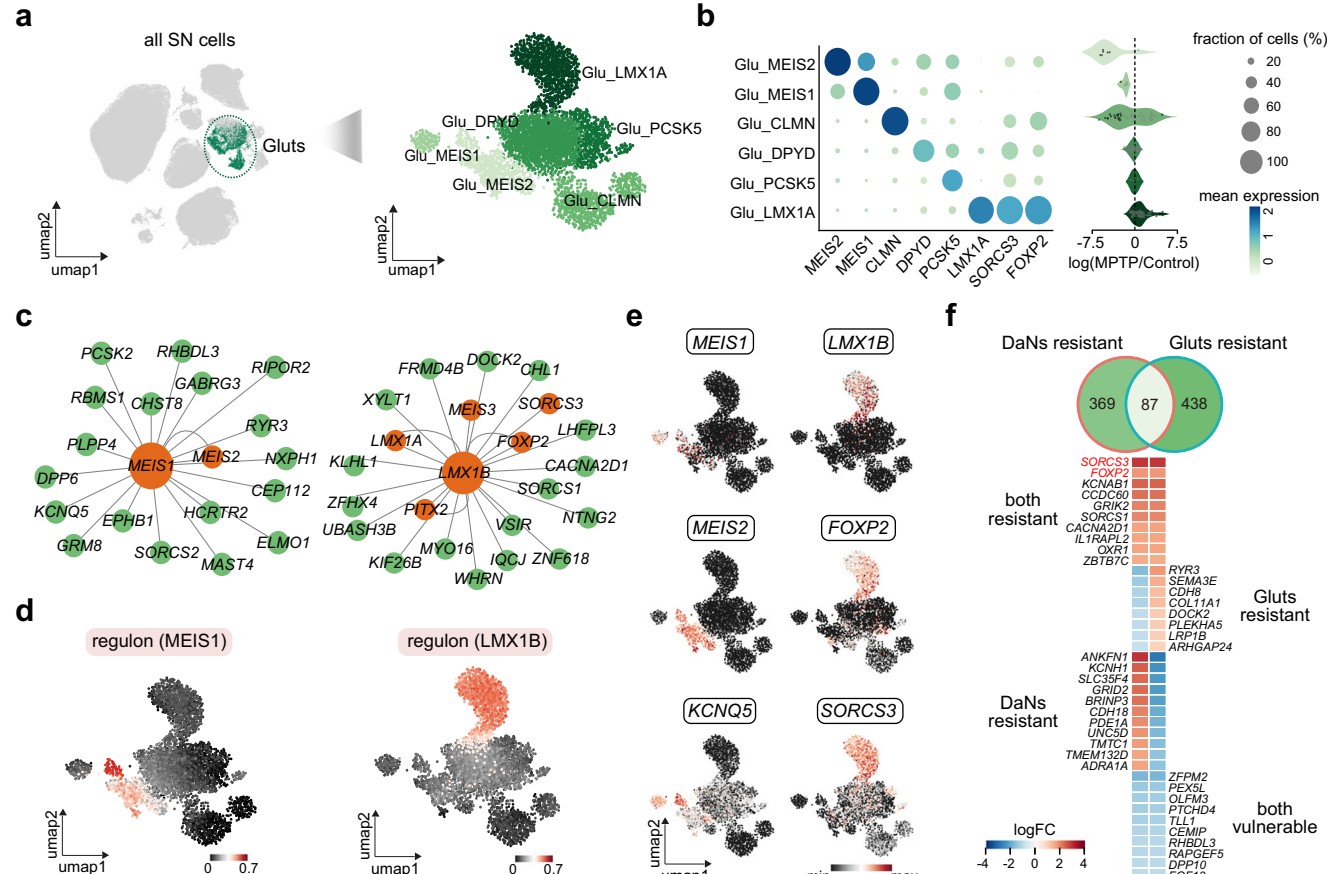

**Fig. 3 | Gradient of glutamatergic excitatory neuronal resilience and the regulatory association with dopaminergic neurons. a** Left: UMAP visualization of all cells in the substantia nigra. Inset shows the glutamatergic excitatory neurons. Right: UMAP visualization of cells colored by the six subtypes identified, with darker color indicating higher neuronal resilience to parkinsonism. **b** Left: dot plot showing expression of selected marker genes across neuron subtypes. Color of the dot represents normalized gene expression, and size represents the percentage of cells expressing a given gene. Right: violin plot showing cell abundance changes between MPTP-treated and control subjects, with the dots representing neighborhoods grouped by cell subtypes. **c** Topological structure of the MEIS1 (left) and LMX1B (right) regulons. **d** UMAP visualizations of glutamatergic excitatory neurons colored by activity scores of MEIS1 (left) and LMX1B (right) regulons. **e** UMAP visualizations of excitatory neurons colored by expression of selected genes within the MEIS1 (left) and LMX1B (right) regulons. **f** Top: Venn diagram demonstrating the overlap of marker genes between PD-resistant dopaminergic neurons and glutamatergic excitatory neurons. Bottom: heatmap showing the fold changes of selected marker genes between PD-resistant and PD-vulnerable neurons for dopaminergic (left column) and glutamatergic excitatory (right column) neurons. Four categories of marker genes are marked in the plot. Source data are provided as a Source Data file.

Its enrichment in the most resilient subtype may suggest a survival strategy involving regulation of nearby neuroinflammation status in DaNs.

Taken together, analysis of the high-quality DaNs in our dataset revealed not only a previously underappreciated transcriptomic heterogeneity but also a gradient of transcriptional susceptibility along with their underlying regulatory units and molecular alterations.

### Transcriptomic signature of neuronal resilience between DaNs and Gluts

We next asked whether and to what extent the neuronal resistance signatures are shared between DaNs and Gluts. To this end, we performed the same analyses on Gluts as with DaNs to resolve their cellular heterogeneity and selective vulnerability in the SN.

We obtained six molecularly defined subtypes of Gluts after reclustering the 4890 neurons, each of which was characterized by a specific marker gene (Fig. 3a, b). Differential abundance analysis showed that two of them, a MEIS1+ subtype and the other MEIS2+ subtype, were susceptible to neuronal loss in the parkinsonian macaques (Fig. 3b). MEIS1 and MEIS2, two members of the MEIS transcription factors, were related to developmental processes including brain development[39]. MEIS1 has a strong genetic association with restless

legs syndrome (RLS)[40], and previous epidemiological studies showed that men with RLS had a higher prevalence of PD[41]. Similarly, recent studies indicated that MEIS2 was capable of regulating the apoptosis process of striatal medium spiny neurons[42]. Confirming this, we located a MEIS1-MEIS2 regulon for the two vulnerable cell subtypes through regulatory network analysis (Fig. 3c, d).

Only one subtype with unique expression of LMX1A and a high regulon score of LMX1B was identified as resilient among all Gluts (Fig. 3b–e). Overexpression of LMX1A and LMX1B was reported to protect the mitochondrial function and prevent DaNs from degeneration in the context of mouse PD models[43]. Interestingly, FOXP2 and SORCS3, the two genes characteristic of the most resilient DaNs subtype in the SN, were also overrepresented in this PD-resistant Gluts subtype, indicating a partially overlapping signature relating to survival of both DaNs and Gluts (Fig. 3b). To explore this in a broader context, we systematically compared the transcriptomic changes between neuronal resilience and vulnerability, and linked the resulting signatures between DaNs and Gluts (Fig. 3f). Among all the vulnerability and resilience markers in DaNs or Gluts, ~20% of them were shared with Gluts or DaNs, respectively. In total, we identified 87 genes with concurrent expression changes between vulnerable and resilient neurons in both Gluts and DaNs, including FOXP2, SORCS3, and a

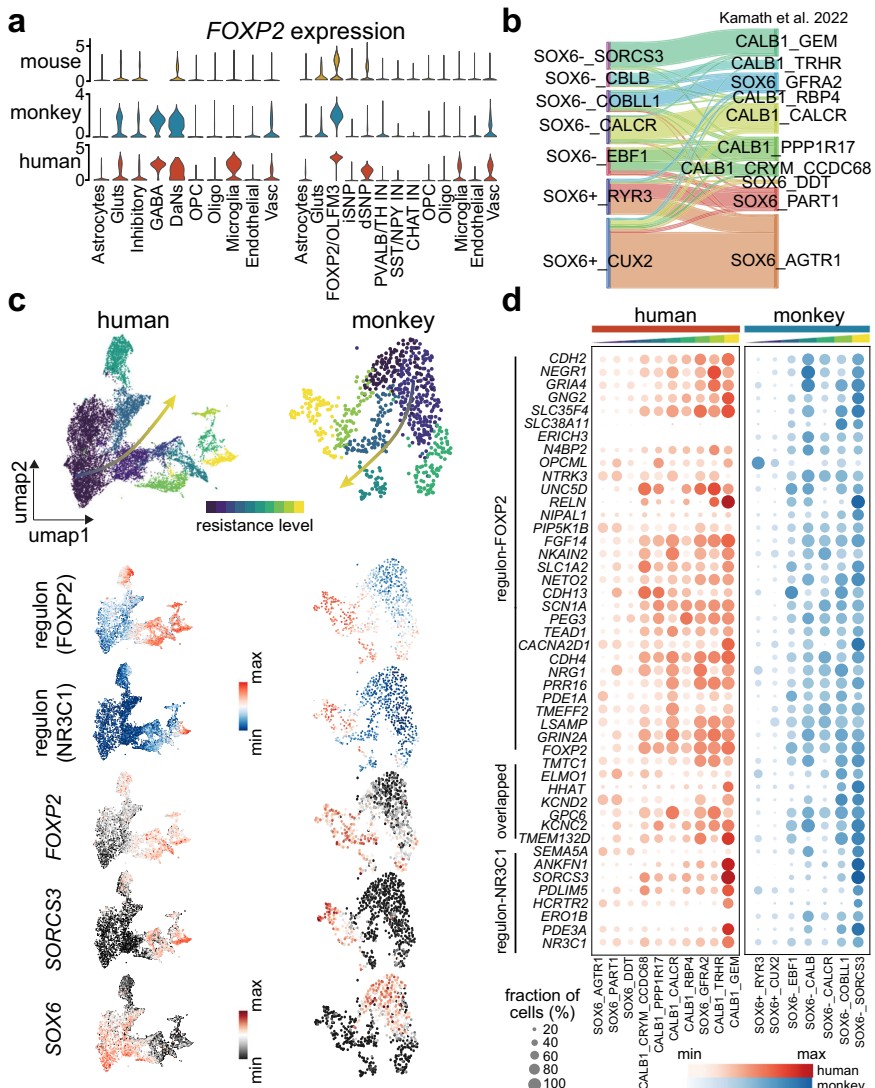

**Fig. 4 | FOXP2 regulon robustly defines dopaminergic neuronal resilience in humans and macaques. a** Violin plot showing expression of *FOXP2* across cell types of substantia nigra (left) and putamen (right) in humans, macaques, and mice. **b** Sankey diagram displaying the correspondence between DaNs subtypes in our data and those from Kamath et al.[10] **c** Top: UMAP visualizations of DaNs in the human (left) and macaque (right). Color gradient denotes the level of neuronal

resilience. Bottom: UMAP visualizations colored by the activity scores of the FOXP2 and NR3C1 regulons, as well as by expression of key vulnerability and resilience markers. **d** Heatmap showing expression distribution of genes in the FOXP2 and NR3C1 regulons across DaNs subtypes of humans (red) and macaques (blue). Source data are provided as a Source Data file.

number of candidate genes such as *LMX1A*, *CHL1*, and *SCN9A* (Fig. 3f and Supplementary Data 5).

**Cross-species comparison reveals a conserved neuronal resistance gradient**

In spite of similar pathological manifestations between our macaque model of parkinsonism and human PD patients, the similarity in their cellular and molecular properties also needs to be interrogated. We therefore utilized previously published single-cell datasets profiling the neuronal and non-neuronal cells in the SN and PT from the mouse[20] and human[10] for comparisons with our model.

We first examined the expression distribution of the resilience marker *FOXP2* across species and cell types. Consistent with a previous study on the primate prefrontal cortex[44], in our data *FOXP2* was also specifically expressed in the human microglia, being absent or sparse in microglia of non-human primates and mice (Fig. 4a). We thus generalized the human-specificity of *FOXP2* in microglia from the neocortex to the nigrostriatal system. Importantly, we found that expression of *FOXP2* in DaNs was primate-specific, as revealed by its

significant enrichment in DaNs of the human and macaque but not the mouse (Fig. 4a), again emphasizing the need for an animal PD model from higher-order species.

At the cell type level, DaNs subtypes in our data demonstrated a good match with the molecular taxonomy of DaNs from the human PD patients and controls[10] (Fig. 4b and Supplementary Fig. 12). For example, CALB1_GEM, the most PD-resistant DaNs subtype in the human SN, was predicted to be the most resilient subtype SOX6⁻_SORCS3 in the macaque SN (Fig. 4b). Similarly, the most vulnerable subtypes in humans, SOX6_AGTR1 and SOX6_PART1, were in general aligned with the two most susceptible subtypes in our data (SOX6⁺_RYR3 and SOX6⁺_CUX2). Given this conservation, we hypothesized that the FOXP2 regulon we identified (including FOXP2 and its target genes) can be exploited to define the gradient of neuronal resilience in DaNs across species. Indeed, the resilient DaNs in both species had higher expression of *FOXP2* and a higher score derived from the FOXP2 regulon compared to those from the vulnerable DaNs (Fig. 4c). We enumerated *FOXP2* and its target genes in Fig. 4d, the vast majority of which showed an ascending expression pattern from

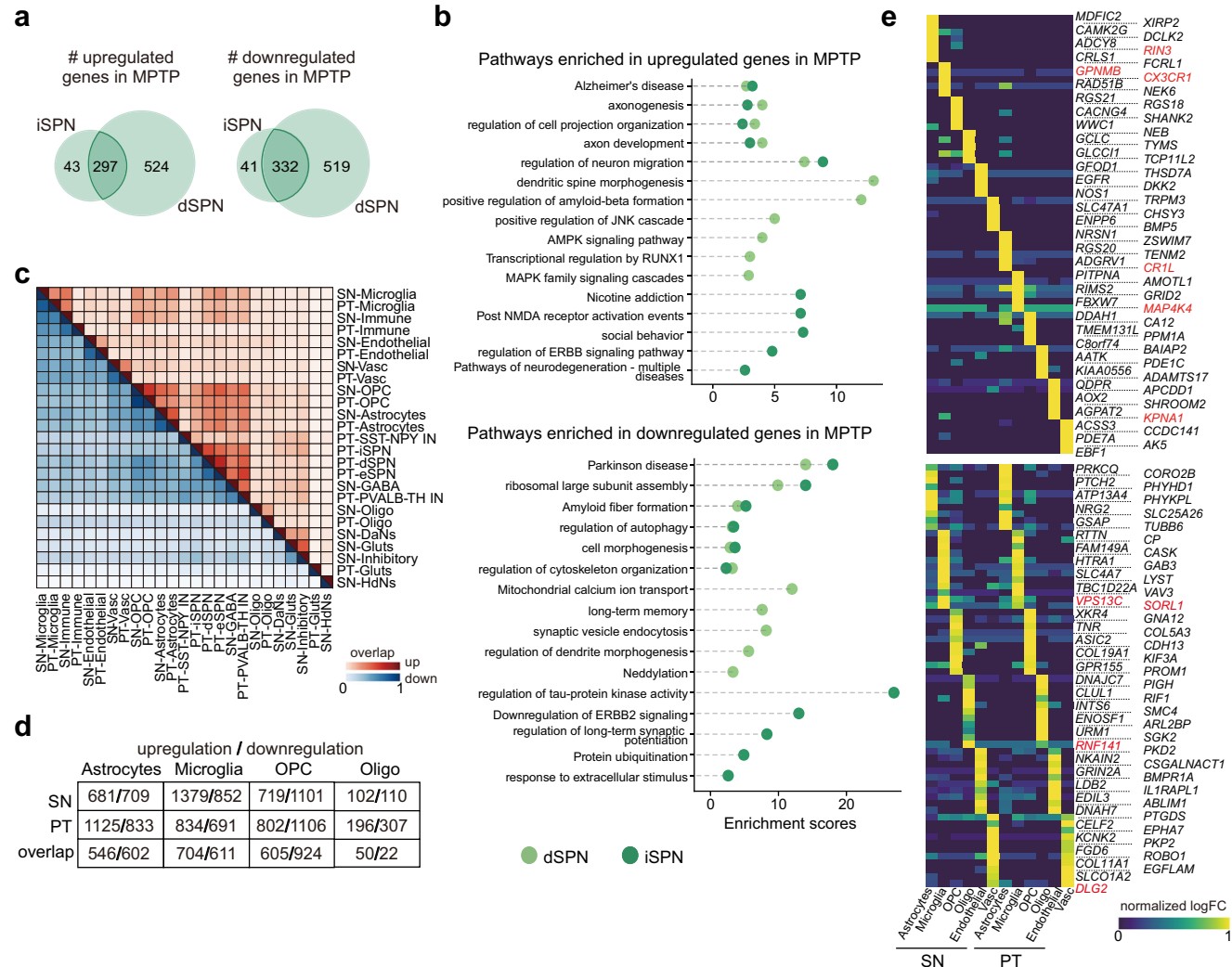

**Fig. 5 | Shared and specific molecular alterations induced by MPTP treatment between putamen and substantia nigra. a** Venn diagram demonstrating the overlap of genes upregulated (left) and downregulated (right) by parkinsonism between iSPN and dSPN. **b** Scatter plot showing the enrichment scores for gene ontology terms that are enriched in upregulated (top) and downregulated (bottom) genes by parkinsonism. Dots are colored by neuronal types (iSPN and dSPN). **c** Heatmap displaying the extent of overlap for upregulated (upper triangle, red) and downregulated (lower triangle, blue) genes by parkinsonism between SN and PT. **d** Table summarizing the numbers of upregulated (red) and downregulated (blue) genes in parkinsonian macaques across four glial types of SN and PT, as well as the numbers of shared genes between the two regions. **e** Top: heatmap showing region-specific expression induced by MPTP treatment in each glial cell type. Bottom: as the top heatmap, but for genes concurrently induced by parkinsonism in both regions. SN substantia nigra, PT putamen. Source data are provided as a Source Data file.

vulnerable to resilient DaNs in both humans and macaques. In addition, for the most resistant DaNs subtype (i.e., SOX6⁻_SORCS3 in macaques and CALB1_GEM in humans), we highlighted a common NR3C1 regulon and its targets such as *SORCS3* (Fig. 4c, d). All these illustrated the similarity of transcriptomic features and neuronal susceptibility in DaNs across primate species, and showcased the effectiveness of the MPTP-induced nonhuman primate model in mirroring natural PD progression.

**Activation of spiny projection neurons and glial cells in the PT**
Loss of dopamine was believed to induce hyperexcitability of spiny projection neurons in the PT, for example, dSPNs and iSPNs[45]. Our sampling strategy of matched SN and PT tissues from the same macaques allowed us to investigate the aberrant nigrostriatal pathways downstream of SN.

Medium spiny neuron is the major neuronal type in the primate striatum as revealed by a previous snRNA-seq study[19]. Consistent with this, we detected two types of SPNs in the PT, including dSPNs and iSPNs[21] (Fig. 1d). They expressed distinct sets of marker genes and constituted the majority of neurons in the macaques PT (Fig. 1f and Supplementary Fig. 3d). During PD, SPNs in the dopamine-denervated striatum exhibited substantial structural changes in their dendritic spines, suggestive of a pathologic state affecting brains beyond the SN region[46,47]. To identify the underlying transcriptional changes, for each type of SPNs we performed differential expression analysis between diseased and healthy subjects. In general, the numbers of differentially expressed genes (DEGs) in dSPNs (upregulated in parkinsonian macaques, 821; downregulated, 851) were shown to exceed those in iSPNs (upregulated, 340; downregulated, 373) (Fig. 5a). Moreover, we found both shared and specific expression alterations between iSPNs and dSPNs.

A number of genes were activated by the PD-like pathogenesis and meanwhile were shared between the two neuronal types. Aberrant dendritic changes and synapse remodeling are common to all SPNs during PD. Consistent with this, these upregulated genes including *ROBO2, PRKG1* and *SLIT3* were revealed by previous studies to

associate with the morphology and transmission of synapses[48–50]. Interestingly, *UBE3A*, whose expression was enriched in many DaNs subclusters, was also upregulated in dSPNs and iSPNs, indicating a widespread protein degenerating process mediated by this gene in PD[51,52]. GO enrichment analysis further showed that "regulation of cell projection organization" and "axonogenesis" were overrepresented by these shared upregulated genes between the two SPNs types (Fig. 5b). With respect to downregulated genes, GO analysis highlighted "Parkinson disease" as highly enriched in the two types of SPNs (Fig. 5b). Cell-type-specific enrichment was also revealed. For example, in iSPNs we observed upregulation of several neurodegeneration-related genes in the parkinsonian status such as *APAF1*, *DNAH9* and *HIP1*. Of note, the term of "regulation of JNK cascade" was enriched exclusively in dSPNs, which was shown by a previous study to have a protective effect on nigrostriatal pathways in injured striatum[53]. On the other hand, PD-downregulated genes in dSPNs were enriched in "mitochondrial calcium ion transport" and "synaptic vesicle endocytosis". For PD-downregulated genes in iSPNs, they were enriched in "regulation of tau-protein kinase activity" and "protein ubiquitination", demonstrating a compromised homeostatic function in the MPTP-treated tissue (Fig. 5b).

Extending this analysis to all cell types in the PT, we found a significant overlap of DEGs between SN and PT (Fig. 5c). This phenomenon was most notable in the glial cells, as almost half (47.5% ± 25%) of genes were concurrently disturbed in the glia of the two regions, indicating a similar response of glia in the SN and PT (Fig. 5c, d). Many of these genes were previously reported to induce glial activation (Supplementary Data 6). For instance, in astrocytes of both SN and PT, a PD-associated gene *PRKN* was upregulated. *PRKN* was involved in mitochondrial dysfunction and its overexpression was reported to trigger a malfunctioning state of astrocytes in PD pathology[54]. Similarly, several other genes such as *UBE3A* and *LRBA* were found to increase their expression in PD astrocytes, both of which were related to immune response regulation and protein ubiquitination (Supplementary Data 6). In microglia, overlap of PD-upregulated genes between SN and PT was smaller, but still comprised a set of immune response activation-related genes such as *CASP6*, *CD4*, *PRKCB*, and *VAV3*, and other genes relating to the apoptotic process and immune response regulation such as *TIAM1*, *ARRB1*, *ELMO1* and *PIK3R1* (Supplementary Data 6). Brain region-specific response, albeit with a relatively smaller magnitude, was also observed for each glial cell type (Supplementary Data 7), such as the preferential upregulation of *GPNMB* in SN microglia and *MAP4K4* in PT microglia, as well as *RIN3* in SN astrocytes and *CR1L* in PT astrocytes (Fig. 5e).

## Discussion

In this study, we focused on the common neurodegenerative disease PD, and provided a single-cell mapping of healthy and pathological cell states in the nigrostriatal area of nonhuman primate species. Besides, our work has three biological and technical strengths: (1) a macaque model of parkinsonism highly representative of important features of human PD pathophysiology; (2) unbiased comparison of parkinsonism and homeostasis with age-, sex-, and environment-controlled design; (3) comprehensive sampling of the nigrostriatal system including the SN and its primary projection target PT. The resulting single-cell atlas contained more than 250,000 cells, providing a valuable resource for understanding cell-type-specific responses to PD and for developing new treatments in therapeutic intervention.

Selective neuronal vulnerability is an important feature of neurodegenerative diseases such as PD in the context of complex brain cytoarchitecture[55,56]. Compared to the post-mortem human brain, our primate model is more informative as it allows easier experimental manipulations to determine the transcriptomic changes as a consequence of PD. For example, this model was shown to mimic

important pathological, cellular and molecular features of human PD. Through this, we found characteristic neurodegeneration patterns of DaNs. Comparison of neuronal loss across DaNs subclusters further showed that MPTP-treated macaques had a similar pattern of neuronal vulnerability as compared to that discovered by a previous study on postmortem human PD tissue[10]. Those results together suggest the advantage and utility of the MPTP-induced primate model in potential mechanistic and therapeutic studies that could contribute to a better understanding of human PD pathology. Specifically, our analysis uncovered a *SOX6*-dependent axis along which neuronal susceptibility can be demarcated in DaNs, similar to the *SOX6-CALB1* axis revealed by a human snRNA-seq study in post-mortem SN[10]. Instead of *CALB1*, in this study we focused on the regulatory unit of *FOXP2* as expression of *FOXP2*, together with the target genes it regulated, were found to be highly restricted to resilient neurons. This possibly represented a conserved survival mechanism given that this pathway was shared by DaNs and Gluts and by nonhuman primates and humans.

*FOXP2*, an important transcription factor involved in nervous system evolution[57], was previously indicated to be associated with birdsong and echolocation in birds or bats[58,59], as well as with speech and language in humans[60]. More recently, several studies have reported its relatedness to neurodegeneration[61,62]. In this study, we found that expression of *FOXP2* and its target genes was overrepresented in resilient DaNs subtypes. Echoing this, a pilot study using living samples detected upregulation of *FOXP2* in the prefrontal cortex of PD patients[62]. This study also revealed a considerable number of differentially expressed genes between living PD patients and postmortem samples, underscoring the necessity of controlling post-mortem effects for human PD specimens. Considering this, a macaque model of parkinsonism, such as the one used in our study, will likely benefit future studies on PD. For example, we observed a good correspondence between DaNs in macaques (including both the vulnerable and resilient neuron types) and those in humans with no new cell states induced by MPTP neurotoxicity.

Linking the neuronal resilience gradient with molecular changes caused by PD shed light on the underlying mechanisms of PD pathophysiology. In most DaNs subtypes, *UBE3A*, whose protein product belongs to the ubiquitin degradation system, upregulated its expression in parkinsonian subjects. Upregulation of *UBE3A* may be protective for DaNs to resolve the protein misfolding caused by oxidative stresses and mitochondrial dysfunctions via activation of the ubiquitin system[63]. As expected, MPTP-causing changes in the vulnerable DaNs (SOX6⁺_RYR3 and SOX6⁺_CUX2) were enriched for GO terms of "positive regulation of apoptotic process" and "positive regulation of cell death". Among PD-resistant DaNs, the most resilient subtype SOX6⁻_SORCS3 was characterized by expression of both *FOXP2* and *SORCS3*, with its PD-induced genes involved in exocytosis microtubule-related terms such as "regulation of exocytosis", "microtubule-based movement" and "microtubule-based transport". Since *RELN* and *TGFB1* were highly expressed in the SOX6⁻_SORCS3 subtype and both of them encode secretory proteins, the mechanism of PD resilience in this subtype may be related with the protective role of their secreted factors.

There are two important considerations for interpreting our results. Firstly, although the vast majority of DaNs subtypes can be paired between the macaque and human, the two vulnerable types in macaques, SOX6⁺_CUX2 and SOX6⁺_RYR3, were matched as a whole to the vulnerable subtypes of SOX6_AGTR1 and SOX6_PART1 in humans. This ambiguity of cell-cell mapping may reflect either the intrinsic human-macaque difference in DaNs subtypes or the value of our data in obtaining a refined cell type division scheme for primate DaNs. Further works will be needed to obtain strictly controlled human specimens for exploring this. Secondly, though *CALB1* was previously regarded as a resilience marker of DaNs with a neuroprotective role in

human PD, its expression between humans and macaques was not fully matched. In comparison to humans, we found a smaller proportion of *CALB1*[+] resilient DaNs in macaques, in keeping with a previous single-cell transcriptomics study in the macaque SN[10]. Similarly, another study comparing the proportions of CALB1[+] DaNs between control and MPTP-induced parkinsonian macaques revealed only a slight increase in CALB1[+]TH[+] cells after MPTP treatment in the ventral tegmental area (VTA)[64], indicating the restricted expression of *CALB1* across PD-resistant DaNs in macaques.

## Methods

### Ethical compliance
All experimental procedures were approved by the Animal Care and Use Committee of Zhongshan Ophthalmic Center, Sun Yat-sen University. The study was performed in accordance with the Principles for the Ethical Treatment of Non-Human Primates.

### MPTP-induced Parkinson's disease macaque model
Adult *Macaca fascicularis* were obtained from the Blooming-Spring Biotechnology Co. Ltd. (Guangdong Province, China). Macaques were tested negative for tuberculosis, B virus, simian T-cell lymphotropic virus, simian immunodeficiency virus, simian type D retroviruses, etc. Eight adult *Macaca fascicularis* were used in this study (Supplementary Data 1). All animals were fed in environmental conditions that were maintained a temperature of $24 \pm 2$ degrees Celsius, a relative humidity of $50 \pm 5\%$, and a 12 h light/12 h dark cycle. MPTP (3 mg; Sigma-Aldrich, St. Louis, MO, USA) was dissolved into a concentration of 0.15 mg/ml and injected singly into the left carotid artery. The dose of MPTP was constant for all macaques regardless of their body weights, as previous studies have shown that the major determinant of direct intra-carotid MPTP dosing is the brain volume that does not correlate with the body weight[65]. The control group was injected with the same dosage of solvent. PD symptoms occur within two weeks of dosing and the Kurlan score gradually increased from 0 and stabilized 3 months after MPTP injection (Supplementary Data 8). Ten months after the MPTP administration, *Macaca fascicularis* were sacrificed. In detail, the macaques were deeply sedated with isoflurane and then euthanized with an overdose of pentobarbital. Next, the macaques were trans-cardially perfused with 2 liters of cold oxygenated artificial cerebrospinal fluid (ACSF, in mM: 125 NaCl, 25 $NaHCO_3$, 1 $NaH_2PO_4$, 2 KCl, 25 D-glucose, 2 $CaCl_2$, 1 $MgCl_2$, 5 sodium L-ascorbate, 3 sodium pyruvate, 0.01 taurine, 2 thiourea, 1 kynurenic acid, 0.1 DL-AP5 and $1 \times 10^{-3}$ tetrodotoxin).

### Single-nucleus suspension preparation
In total, six macaque monkeys (6–18 years old, three males and three females; two parkinsonian subjects and four control subjects, one for single-cell and five for single-nucleus RNA sequencing) were used. After macaques were sacrificed (see the above section), the skulls were removed and the whole brain was separated carefully. The SN and PT part of the brain was isolated and referred to as the "A Combined MRI and Histology Atlas of the Rhesus Monkey Brain"[66]. The isolation protocol mainly comes from the published 10x Genomics protocol for nucleus isolation (https://www.10xgenomics.com/support/single-cell-gene-expression/documentation/steps/sample-prep/isolation-of-nuclei-for-single-cell-rna-sequencing-and-tissues-for-single-cell-rna-sequencing). In brief, the tissue was lysed via a chilled lysis kit (Sigma, no. NUC101-1KIT). The suspension was then filtered with cell strainer and washed impurities by centrifugation with different concentrations of iodixanol. Nuclei-pellets were resuspended with a resuspension buffer(RB, 1× phosphate-buffered saline, 1% bovine serum albumin and 0.2 U/μl RNase inhibitor). Nuclei-pellets were suspended in the DAPI solution and incubated for inspection under the microscope. Sample suspension with intact nuclei were collected and the nuclei concentration was adjusted prior to loading to the sequencer.

### Single-cell suspension preparation
Live cell extraction experiments were performed as previously described[67]. First, we placed the tissue in the cold N-methyl-d-glucamine solution (NMDG, in mM: 93 NMDG, 30 $NaHCO_3$, 1.2 $NaH_2PO_4$, 2.5 KCl, 20 HEPES, 25 D-Glucose, 5 Sodium Ascorbate, 2 Thiourea, 3 Sodium Pyruvate, 10 $MgSO_4$, 0.5 $CaCl_2$, 93 HCl) and then cut the tissue into 300-μm thick slices by vibratome (Leica VT VT1200S). Next, we separated the target tissue under a microscope and cut it into small pieces. The dissociation solution includes 40 U/ml of papain (Worthington Biochemical), 20 mg/ml of collagenase type II (Worthington Biochemical), 0.25% trypsin-EDTA (Gibco), 5 mg/ml of DNase I (Roche) in ACSF warmed at 34 °C for 10 min. In order to obtain high cell yields of viable populations, we also added neuroprotective agents to the dissociation solution, containing kynurenic acid (1 mM), NBQX (20 μM), DL-AP5 (100 μM) and TTX (1 μM). The small pieces were transferred to dissociation solution, incubated for 60 min at 34 °C, and simultaneously bubbled with a carbogen gas (95% $O_2$ and 5% $CO_2$). After incubation, tissue was gently blown with a pipette several times, the tube was briefly spun, and the supernatant was gently removed. The pieces were resuspended with 10% FBS (Gibco) in ACSF and filtered through a 70-μm cell strainer. The debris and dead cells were removed using Myelin Removal Beads II (Miltenyi) and Dead Cell Removal Kit (Miltenyi) and then washed two times with 0.04% PBS-BSA. Cells were counted (Bio-Rad, TC20) after visual inspection and diluted to desired concentrations for scRNA-seq. All experimental processes were operated on ice or at 4 °C except for enzymatic digestion. Samples with cell viability of higher 90% were used for the following library and sequencing.

### Immunolabeling and FANS for enrichment of dopaminergic nuclei
Two male macaque monkeys (one parkinsonian and one control subject, Supplementary Data 1) were used for FANS. Immunolabeling of *Macaca fascicularis* nuclei was described by a previous study[10]. In brief, RB was added to the resuspension of nuclei. Antibodies NR4A2-A647 (Santa Cruz, no. sc-376984 AF647) and DAPI (Thermo Fisher, no. 62248) were added to the resuspension at concentrations of 1:350 and 1:1000, respectively. Samples were incubated at 4 °C for 1 h under a lightproof condition, and then centrifuged at $150 \times g$ for 10 min and the supernatant was gently aspirated. Next, the concentrated nuclei were resuspended by 1 ml RB and filtered with a 70 μm filter. Flow-sorting was described by a previous study[10]. In brief, we used a 1.5 ml tube coated with RB to collect sorted nuclei. Subsequently, the filtered single nucleus suspension was transferred into a 5 ml tube (Corning Falcon, no. 352063) and sorted by flow cytometer (BD, FACSAria Fusion). The nuclei were sorted under the model of "purity" with a flow rate of 6.0 at 4 °C. The data were collected by the BD FACSDiva software (v9.0). For NR4A2[+] selection, a DAPI versus 647 gating was established by selecting the 16.8% most highly fluorescent NR4A2[+] nuclei. Flow cytometry data were analyzed, processed and visualized with the FlowJo software (v10.3).

### Library preparation and sequencing
After adjusting the nucleus concentration, the suspension was loaded to the sequencer. Nuclei were processed using the Chromium Next GEM Single Cell 3'Kit v3.1 to generate the cDNA libraries. Quality of cDNA was assessed using the Agilent 2100 Bioanalyzer System, and sequencing was performed on Illumina NovaSeq6000-S2.

### Data preprocessing
Gene expression count matrices were generated by mapping the sequenced reads using Cell Ranger (10x Genomics) v.6.0 with the default parameters, which took in account both the exonic and intronic reads during read collapsing and UMI counting. The *Macaca fascicularis* genome (version 5.0)[68] was downloaded from Ensembl and

formatted using Cell Ranger version 6.0 with default parameters. For quality control, cells with higher than 10% (single nucleus) or 30% (single cell) mitochondrial contents and with fewer than 200 features/genes were filtered out from the downstream analysis. Features that were expressed in fewer than three cells were also filtered out from the analysis. Doublet scores were calculated by Scrublet[69] (version 0.2.1) with default parameters and the predicted doublet cells were filtered out.

## Cell clustering and annotation

Scanpy (version 1.8.2) was used for data analysis following the official tutorials[70]. After quality control, we performed data normalization, log-transformation, highly variable gene selection, data scaling, and principal component analysis. For data normalization, normalized counts(NC) were calculated as transcripts per million (TPM) divided by 100 (i.e., with a scaling factor of 10,000), and then $\log_2(NC+1)$-transformed. Highly variable genes were selected based on the log-transformed data and principal component analysis was conducted with these genes after expression scaling. Then the package Harmony (version 1.0)[71] was used to correct batch confounders from samples and generate a corrected latent space. Uniform Manifold Approximation and Projection (UMAP) was used for two-dimensional reduction and visualization. Cells were clustered and further annotated based on the expression of canonical marker genes. Specifically, *SLC17A6*, *GAD1*, *MOBP*, *VCAN*, *CSF1R*, *DCN*, *AQP4*, *HDC*, *SLC6A3*, and *FLT1* were used to assign cells to one of the following main lineages: glutamatergic excitatory neurons, GABAergic inhibitory neurons, oligodendrocytes, oligodendrocyte progenitor cells, microglia, vascular leptomeningeal cell, astrocytes, histaminergic neuron, dopaminergic neurons, and endothelial cells. Cells from each cell category were further extracted and re-clustered to achieve higher-resolution with the same procedure as mentioned above. Integration of single-cell and single-nucleus datasets was performed by scVI[72] following the official tutorials.

## Differential expression analysis

Differentially expressed genes (DEGs) for each cell population were obtained by using the Seurat FindMarkers function by default parameters. The top 400 (600) genes from each cell type (each cell category) were selected for Gene Ontology (GO) and Kyoto Encyclopedia of Genes and Genomes (KEGG) enrichment analysis using Metascape[73]. Representative terms shown in the figures were selected within the top 100 terms with the cumulative hypergeometric $P < 0.05$.

## Differential abundance analysis between parkinsonian and control subjects

To identify differentially abundant cell populations in association with parkinsonism, we used the milopy package under the standard workflow, which tested the differential abundance based on cell distributions within the KNN graphs[24]. In brief, we first built a KNN graph using the Harmony-corrected PCA space and then assigned cells to their neighborhoods with the milopy function "make_hoods". Next, we counted cells in each "nhood" with the function "count_nhoods" and tested the differential abundance between MPTP and control subjects using the function "DA_nhoods".

## Cell type label transfer

Human and macaque genes were matched according to the uniquely mapped orthologous gene list from Ensembl. The human DaNs dataset was downloaded from the Gene Expression Omnibus (GEO) under the accession code (GSE178265)[10]. Cell type label transfer from the human to macaque using logistic regression-based algorithm was performed with SCCAF (version 1.0.0) packages[74]. The $\log_2(NC+1)$-transformed values were used for further analyses. The classifiers were trained with 38 different parameters (here, testing the range of the numbers of HVGs from 200 to

4000). The selected genes were used for principal component analysis and then Harmony-corrected latent space was used for training and prediction. The transferred labels were based on the majority voting among the 38 predictions from the classifiers[75].

## RNA velocity analysis

RNA velocity (version 0.17.15)[76] and scVelo (version 0.2.2)[77] were used for RNA velocity analysis and "run 10x" options were used to obtain velocity loom files from 10x Cell Ranger results. Then the obtained loom files were merged with Scanpy AnnData file by scVelo. RNA velocity was calculated using the default parameters of scVelo. The moments are computed for each cell across its nearest neighbors, where 30 principal components and 30 neighbors were used. To recover the full splicing kinetics genes, recover_dynamics function was used. The velocities were estimated by using the velocity function with the dynamical mode. The velocity graph was calculated by the velocity_graph function with default parameters.

## Robust cell type decomposition analysis

To explore the spatial distribution of DaNs subtypes, we assigned cell types from single-cell and single-nucleus dataset to previously published Slide-seqV2 pucks by using the robust cell type decomposition (RCTD) package under the standard workflow[78]. The gene set used in RCTD was created by choosing the common genes in quality controlled single-cell/nucleus data and Slide-seqV2 detected genes. The mapping was performed with the "doublet" model, and "reject" beads were filtered out for downstream analyses.

## Identification of regulatory networks

To identify the differentially regulated regulons associated with specific subtypes of DaNs and Gluts, we used pySCENIC[79] with defaults settings from the SCENIC vignette (https://pyscenic.readthedocs.io/en/latest/). Briefly, we ran a correlations analysis using GENIE3 based on a human transcription factor list. The indirect targets were pruned using cis-regulatory motif discovery and the enrichment scores of the regulons were quantified using AUCell, followed by visualization on the UMAP embeddings. The topological graphs were visualized by Cytoscape (version 3.9.1).

## Immunohistochemistry

Adult macaque monkeys were deeply anesthetized and then euthanized with an overdose of pentobarbital. The SN part of the midbrain was dissected, fixed with 4% PFA for 6 h, and cryoprotected in 15% sucrose at 4 °C for 24 h, followed by cryoprotection in 30% sucrose at 4 °C for 24 h. The tissue samples were frozen in optimal cutting temperature compound (OCT; Tissue-Tek) at −80 °C, sectioned at 30 μm on a cryostat microtome (Leica CM1950). Sections were first rinsed in PBS, incubated for 30 min in 0.5% Triton X-100 (Sigma-Aldrich), and then incubated for 2 h in 5% donkey serum (Vector Laboratories) as blocking solution, followed by incubation with primary antibodies overnight. Incubated tissues were washed by PBST for three times and then incubated for 2 h at room temperature with the secondary antibodies. The antibodies used included anti-Tyrosine-Hydroxylase (AF488, Millipore, MAB318-AF488, 1:500), anti-FOXP2 (rabbit, Abcam, ab16046, 1:1000), anti-KCNJ6 (rabbit, Alomone Labs, APC-006, 1:1000), anti-SORCS3 (rat, R&D, MAB3067, 1:100), anti-glial fibrillary acidic protein (chicken, Millipore, AB5541, 1:2000), anti-IBA1 (rabbit, Wako, 019-19741, 1:500), anti-OLIGO2 (rabbit, Millipore, AB9610, 1:500). Next, the sections were mounted with DAPI (Sigma) and coverslipped. Slides of images were obtained with a LSM880 Zeiss confocal microscope.

## Reporting summary

Further information on research design is available in the Nature Portfolio Reporting Summary linked to this article.

## Data availability

The single-cell RNA-sequencing data generated in this study have been deposited in the EMBL-EBI database under accession code E-MTAB-13437. Public datasets utilized in this study are available in the Gene Expression Omnibus database under accession codes GSE178265, GSE152058, and GSE116470. All data supporting the findings of this study are provided within the paper and its Supplementary Information. Source data are provided with this paper.

## Code availability

All custom code used in this work is available at the following GitHub repository: https://github.com/leitang607/Parkinson_disease.

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

## Acknowledgements

This research is supported by research grants from the Natural Science Foundation of China (81961128021, 82371095), the National Key R&D Program of China (2022YEF0203200), the Guangdong Provincial Key R&D Programs (2018B030335001, 2018B030337001), the Science and Technology Program of Guangzhou (202007030011, 202007030010, 202007030001) to S.L.; the Guangdong Provincial Key Area R&D Program (Grant No. 2023B1111050004), the National Natural Science Foundation of China (No. 82271095), the National Key R&D Program of China (2018YFA0108300) to X.L.; the STI2030-Major Projects (2021ZD0200900) of China, the National Key Research and Development Project of China (2018YFA0108503), the Hainan Key Research and Development Project (ZDYF2021SHFZ049) to F.Y., the National Key R&D Program of China (2021ZD0200103) to R.L.; the National Natural Science Foundation of China (No.82101329) to W.Y.

## Author contributions

S.L., C.X. and X.L. conceived and supervised the project. L.T., N.X., M.H., W.Y. and X.S. performed the single-cell RNA-sequencing experiments. R.L., W.Y., F.Y., Y.S. and L.T. performed animal experiments. L.T., C.X. and Z.-Z.H. analyzed the data. N.X., L.T., M.H., M.S. and X.S. performed the immunofluorescence staining. S.L., C.X., X.L., Y.L. and L.T. wrote the paper with the input from all authors.

## Competing interests

The authors declare no competing interests.
