## [Peer Review File · Nature Communications]

A primate nigrostriatal atlas of neuronal vulnerability and resilience in a model of Parkinson's diseaseREVIEWER COMMENTS

Reviewer #1 (Remarks to the Author):

In their manuscript, Tang et al performed sn- and sc-RNA-seq on MPTP-treated and control macaques to identify pathways associated with cell vulnerability and resilience. They find that a FOXP2-centered regulatory pathway, shared between DA and excitatory neurons, is associated with neural resilience. They also report microglial activation in SN and PT. Overall, the data appear to be of high quality (though some mixing of sc- and sn-RNA-seq is confusing, and mentioned below). With proper model validation (see comment #1 below), a sizable community of researchers focused on PD will value having single cell -omics data from MPTP-treated monkeys.

A main question here is of the novelty and importance of the findings. An extensive atlas of both human and macaque substantia nigra was published last year (Kamath et al PMID 35513515), which aligned macaque and human cell types (as well as other mammalian species), and identified the neuronal populations most vulnerable to degeneration in actual PD patients. That published work also identified an activated microglial state associated with PD in humans.

The introduction of this manuscript states that the ability to MPTP-treat monkeys enables one to avoid postmortem effects of human tissue, and to assay cell states earlier in disease progression. However, the MPTP treatment was performed 10 months prior to sacrifice, which strikes me as a very long time. Indeed, the present work seems to mostly support the idea that MPTP treatment in monkeys generally recapitulates the cell type vulnerabilities and glial activation signatures found in human subjects. It does not find a state specific to MPTP treatment, or perform a temporal trajectory analysis that might be suggested from how the authors advertise the advantages of the MPTP model.

Major comments:

1. A major missing element in this work is a lack of behavioral or pathological characterization of the MPTP-treated monkeys. How did the authors assess if a single dose MPTP injection can produce stable parkinsonian symptoms? A Kurlan score is briefly mentioned in the methods, but no data are presented. The details must be mentioned and data shown. Stable maintenance of PD symptoms is necessary for these types of studies. Some monkeys may show recovery by compensatory mechanisms (e.g. serotonin pathway). Previous studies suggested some guidelines to assess the stability of symptoms for 8 weeks after the MPTP injection. It is also essential to show basic pathological analyses of the analyzed tissues. Is there obvious gliosis in SN or PT? Can a rough estimate of the DA loss be obtained? Are there any other brain regions that are noticeably affected?
2. The authors repeatedly refer to the macaques that received MPTP as "PD." This is misleading terminology. Injecting a monkey with a toxin is not the same as Parkinson's disease, which is a human disease. The text and figures should be updated to properly label these monkeys.

3. One sample (control FM-1128) went through single-cell RNA-seq and I assume it was not included in any of the analyses. Can the authors mention the reason, and what differences the scRNA-seq emerged compared with the snRNA-seq data? For example, I see that the proportion of oligodendrocytes is much higher in the snRNA-seq samples compared with scRNA-seq. If it is because of using Myelin Removal Beads II in single-cell suspension preparation compared to single-nucleus suspension preparation, it should be clearly mentioned when describing the proportion of cell-types and its comparison with other studies.

4. “At the cell type level, DaNs subtypes in our data demonstrated a good match with the molecular taxonomy of DaNs from the human PD patients and controls”.

“The two vulnerable types in macaques, SOX6+_CUX2 and SOX6+_RYP3, were matched as whole to the vulnerable subtypes of SOX6_AGTR1 and SOX6_PART1 in humans.”

The above-mentioned sentences are ambiguous. How did the authors match human data with macaque data? If they used label transfer, there need to be confusion matrices and/or river plots, along with joint marker plots, to quantitatively demonstrate the quality of the alignment.

5. How many dopaminergic neurons were detected in cases and controls separately? A total of 730 dopaminergic neurons probably may not be sufficient to recluster them and investigate the transcriptional heterogeneity of dopaminergic neurons. This may be the reason that no PD specific sub-clusters were found. I suggest increasing the number of dopaminergic neurons by including more samples and/or using an enrichment method such as FACS.

6. In Supplementary Figure 3, the authors showed the proportion of cell-types in cases vs. controls. It is highly recommended to test (e.g. permutation test) whether the proportional alterations are statistically significant or not.

7. There is some previous evidence regarding the correlation between FOXP2 and neurodegeneration (PMID: 28798667). There is a recently published study on the human prefrontal cortex of PD subjects reporting FOXP2 as an important transcriptomic change in PD (PMID: 36952070). The authors can use the papers when explaining the potential role of FOXP2.

8. Finding MEIS1 as one of the markers of susceptible excitatory neuron subtypes could be interesting. One possible explanation could be that MEIS1 is one of the known genes in restless legs syndrome (RLS), a disease with disruption in dopamine levels. Several epidemiological studies reported that RLS risk is higher among PD patients and PD is more commonly seen in patients with RLS compared to the general population (PMID: 31004074).

9. The authors mention that CALB1 is not a marker of the resilient neurons, but many papers studying

DA neurons in primates have seen the CALB1 gradient, including single cell data from PMID 35513515 in macaque. Can the authors explain the discrepancy with their data? Is it due to low numbers of cells, or perhaps some differences in the genome reference or alignment pipelines used in the different studies?

Minor comments:

1. “we also found four PD risk genes (SH3GL2, LRRK2, STAB1, BST1), and an Alzheimer’s disease risk gene (CR1), all of which were highly expressed in these vulnerable DaNs subclusters”.

Did the authors analyze all PD risk genes in both vulnerable and resistant subtypes? If so, the list of genes should be mentioned in a supplementary table.

2. I assume that the weight of the monkeys was not considered when determining the dose of MPTP injection? Can the authors justify this in the methods?

3. A total number of regulons specific to the dopaminergic neuron subtypes identified in this dataset should be mentioned.

4. It is mentioned that the STAB1 gene is a PD-risk gene. Stabilin 1 (STAB1) is not a PD risk gene and apparently, it should be SATB1 (PMID: 31701892).

5. The authors mentioned that “They highly expressed KCNJ6, a marker gene relating to vulnerability in PD24”. However, according to the conclusion of reference 24, GIRK2 (KCNJ6) is unlikely to be a marker to distinguish between dorsal and ventral tier SNpc dopaminergic neurons and is not correlated with differential vulnerability. The authors should either remove it or cite a different paper.

6. The authors should be more consistent in using terms. For example, 10x (10X), Cell Ranger (Cellranger), etc.

7. A full list of antibodies and their characteristics must be included as Supplementary Material.

Typographic:

- Line 102 “Board”

- Line 121 “in partial”

- Line 315 “as as”

- Line 413 “ell population”

-

Reviewer #2 (Remarks to the Author):

In this paper, the authors used single-cell nucleus transcriptomic profiling technology to assess the molecular heterogeneity of midbrain dopaminergic neurons and striatal projection neurons in the putamen of rhesus macaques. To determine if specific genes expression is altered in parkinsonian state, the ventral midbrain and putamen of both control and acutely MPTP-treated parkinsonian monkeys were examined, with the ultimate goal of determining if the expression changes of specific transcriptomes confer neuroprotection or susceptibility of various subtypes of dopamine and striatal neurons to MPTP neurotoxicity.

Although similar approaches were used to associate molecular heterogeneity and sensitivity of midbrain DAergic neurons to PD-induced degeneration in humans and mice, the present study complements the published literature with the addition of data from rhesus monkeys. Regarding the striatal data, the present results are in part confirmatory to those recently published by He et al (Current Biology, 2021; 31:5473), but the information about striatal changes in gene expression in the putamen of MPTP-treated monkeys is new.

Overall, this is a timely study that uses cutting-edge transcriptomic approaches. Despite the fact that a significant amount of data is confirmatory of the published literature, part of the content of this paper extends these previous findings and further strengthens that the molecular heterogeneity of midbrain DA neurons may contribute to their selective resistance or sensitivity to MPTP-induced neurotoxicity. However, there are significant limitations that must be addressed:

1) The validity of the animal model used to assess pathogenesis of midbrain dopamine neurons.

Although MPTP-induced toxicity of dopamine cells follows a pattern reminiscent of what has been observed in human PD, the acute nature of the intracarotid MPTP monkey model used in this study has some limitations. A chronic low dose MPTP-treated monkey model would have been more valuable. Another major concern about the MPTP monkey model is the fact that MPTP-treated monkeys do not exhibit Lewy bodies which is an important shortcoming of this when this model (or any other neurotoxin-based model of PD) is used to assess dopamine cells pathogenesis. These concerns limit the significance of the present results as they relate to the human PD condition.

2) In the ventral midbrain, the authors compare transcriptomic data collected from different types of DA cells and Excitatory neurons. It is not clear based on the information provided what population of neurons account for these excitatory cells. In the SNc and VTA, subsets of neurons co-express DA and vGluT2. Are these the neurons the authors refer to or is it a population of non-dopaminergic glutamatergic neurons. Please, provide more information about the transmitter co-expression phenotype of this neuronal subpopulation.

3) Although of potential interest the striatal data are in large part confirmatory of recent findings published by He et al in Current Biology. The authors must cite and discuss the content of this paper in detail.

4) In lines 164-165, the authors state: "The vulnerable DaNs subtypes upregulated an extensively overlapped set of genes, indicating a similar response induced by PD pathology". This statement is confusing because the so-called vulnerable DANs in the SN of MPTP treated monkeys have died and are

not part of the tissue analyzed from these monkeys. If some neurons remain in the SN of these parkinsonian animals, shouldn't they be categorized as resilient (instead of vulnerable) to MPTP neurotoxicity? This is a limitation of the animal model used in the study.

Minor corrections:

- Avoid using the terminology "PD macaques" when referring to the MPTP monkey model because these animals do not have PD. For this reason, the MPTP monkey model is often considered as a model of parkinsonism instead of PD.
- Lines 15-16: Wording of some statements must be improved. For instance, in the abstract the authors say ".....progressive loss of neurons in the nigrostriatal system including the substantia nigra (SN) and putamen (PT)". This statements sounds as if bot the SN and PT contains nigrostriatal neurons
- Line 20: "Our PD model recapitulates the pathologic features in nature PD". This over-statement is incorrect. The MPTP model recapitulates "some" of the pathologic features in nature PD.
- Line 24: "excitatory neurons" If these are DA/vGluT2 neurons, these cells should be names accordingly to avoid confusion. As currently presented, they are considered as a subset of non-DA glutamatergic neurons in the SN. Is it the case??

Reviewer #3 (Remarks to the Author):

This is a study of the transcriptomic profile of nigral and putamenal cells in non-human primates using the MPTP model of Parkinson's disease. To determine transcript changes, particularly underlying cell vulnerability, 10X genomics chromium platform for sc and snRNAseq was applied comparing samples from parkinsonian and control macaques. Data were analyzed with established methodologies. Most significant results are that SOX6+ expression was associated with DA neuron vulnerability while SOX6- cells especially those expressing FOXP2 and SORCS3 corresponded with higher resistance. The available human data (SOX6-CALB1) support these results. The topographic distribution of the profiled vulnerable and resilient neurons was correlated with the typical distribution of PD pathology between SNc ventral and dorsal tiers. Analyses of the regulated pathways also showed interesting transcript regulation, including support for the presence of inflammation. The study also identified other genes with different expression patterns in vulnerable versus resistant cells.

As expected from this type of descriptive study, comparisons across species and cell types are important. Results showed some misalignments in comparisons with human data, for example in microglia, but there were large similarities in the nigral DA neurons. Overall, these data can contribute to characterizing mechanisms of cell death and survival in PD.

Some concerns come from the following points:

The activation of immune responses (microglia and astrocytes) in SNc, and the striatum as well, are

more difficult to interpret in relation to PD considering that samples derived from an acute toxin model.

There are many transcriptomic changes described in the putamen, but this section is not well organized. Genes related to synaptic transmission, neurodegeneration, inflammation, etc. are intermingled making it difficult to extract the most relevant information. In addition, some data are difficult to put in the context of PD, such as the regulation in striatal neurons of genes related to apoptosis.

The discussion suffers of miscalibration of data generated in the primate MPTP model of PD. Although the model can be said that is, quote, 'representative of PD pathophysiology', clearly it is not representative of PD pathogenesis, which is most relevant to a transcriptomic analysis. One recognizes that the vulnerability of certain neurons, as an inherent condition, may be determined by some shared genes between human and non-human primates independent of the insult. But having a different causative mechanism plays a major role in the determinants of cell vulnerability to degeneration and death, and likely because of different gene regulation. At a minimum, studies to determine if the transcript changes in this animal model are also found in the human disease are necessary for validation of these data.

Comments related to the functional diversity of the putamen, linking dopamine neuron subtype connections to clinical features in patients are not only beyond the scope of this study but also not well founded.

Stella M. Papa

Please find a point-by-point response to the reviewers' comments below, and all changes in the main text have been highlighted in blue.

Comments	Answers
Reviewer #1	
In their manuscript, Tang et al performed sn- and sc-RNA-seq on MPTP-treated and control macaques to identify pathways associated with cell vulnerability and resilience. They find that a FOXP2-centered regulatory pathway, shared between DA and excitatory neurons, is associated with neural resilience. They also report microglial activation in SN and PT. Overall, the data appear to be of high quality (though some mixing of sc- and sn-RNA-seq is confusing, and mentioned below). With proper model validation (see comment #1 below), a sizable community of researchers focused on PD will value having single cell-omics data from MPTP-treated monkeys.	We thank the reviewer for considering our data as high-quality and valuable to the community, as well as for raising concerns in MPTP model validation and cell versus nucleus data mixing. Please find our response to these concerns, together with extra analyses, new figures, tables, and revised text, in the point-by-point response below.
1. A main question here is of the novelty and importance of the findings. An extensive atlas of both human and macaque substantia nigra was published last year (Kamath et al PMID 35513515), which aligned macaque and human cell types (as well as other mammalian species), and identified the neuronal	We thank the reviewer for this point. In comparison to Kamath et al. 2022 which profiled ~44k healthy cells in macaque substantia nigra, our single-cell data comprises ~250k macaque cells under both health (30.3%) and parkinsonism (69.7%), and covers an additional nigrostriatal region putamen. This allows us to directly assess the effect of parkinsonism at scale, and to investigate the influence of MPTP-induced parkinsonism across the nigrostriatal system. By analyzing this dataset, we uncovered a robust axis of neuronal vulnerability and

populations most vulnerable to degeneration in actual PD patients. That published work also identified an activated microglial state associated with PD in humans.	resistance in the substantia nigra between humans and macaques, showing, for the first time, the effectiveness of the MPTP-treated macaque model in mirroring a number of human PD features at the cellular and molecular levels. In addition, we have provided novel insights into: 1) a previously uncovered HDC⁺ neuronal cluster in the substantia nigra; 2) a proliferating microglial subtype possibly responsible for generation of pathological microglia (i.e., the GPNMB⁺ and VAV3⁺ subtypes); 3) identification of vulnerability and resilience markers including shared (e.g., SOX6, ALDH1A1) and novel (e.g., FOXP2, SORCS3) markers when compared to current human-focussed studies, as well as immunoprofiling of selected markers not validated previously. Of note, we revealed for the first time that the FOXP2-centered pathway is activated in resistant DaNs subclusters in both humans and monkeys. 4) the underappreciated link between dopaminergic and glutamatergic excitatory neurons, including partially shared survival mechanisms and neuronal subtype-specific molecular alterations under PD; 5) concurrently perturbed nigrostriatal pathways in the putamen downstream of substantia nigra in our model of parkinsonism. In summary, using the MPTP-treated model followed by matched sampling of the substantia nigra and putamen, we have complemented existing midbrain single-cell datasets with healthy and pathological cell states in the nigrostriatal area of nonhuman primate species, and shed light on both known and underexplored mechanisms of PD pathogenesis. To underscore the value of our dataset and the derived findings, we have now expanded the text in the Discussion section to summarize the above points (lines 319-325, lines 329-341, lines 345-351 in the revised manuscript).
2. The introduction of this manuscript states that the ability to MPTP-treat monkeys enables one to avoid postmortem effects of human tissue, and to assay cell states earlier in disease progression. However, the MPTP treatment was performed 10 months prior to	The macaques used in our study were sacrificed 10 months after the MPTP administration. With this model, we observed a stable and obvious parkinsonism in macaques treated with single carotid injection of MPTP. The curve of Kurlan score over time (Figure R1) showed that the score escalated within 20 days post MPTP injection and was stabilized thereafter. We observed very stable clinical symptoms even 210 days after MPTP administration and verified the clinical signs and symptoms, which indicated the success of parkinsonism modeling in monkeys.

sacrifice, which strikes me as a very long time. Indeed, the present work seems to mostly support the idea that MPTP treatment in monkeys generally recapitulates the cell type vulnerabilities and glial activation signatures found in human subjects. It does not find a state specific to MPTP treatment, or perform a temporal trajectory analysis that might be suggested from how the authors advertise the advantages of the MPTP model.

Single-cell analysis also indicated that the gene expression and cell rate changes in SNc are similar to the human PD progression.

Figure R1. Loss of dopaminergic neurons in the macaque model of parkinsonism. **a** Temporal pattern of the Kurlan scores after MPTP injection. Shown are patterns for two MPTP-treated macaques sampled across 17 time points. Kurlan scores after day 210 became stable and were not recorded. **b** IHC images showing TH stainings in the substantia nigra of control and MPTP-treated macaques. Scale bar, 500 μ m. Box plot to the right displays the numbers of TH-positive cells per slice between control and parkinsonian macaques, with the box frames extended from the lower quartile, median, to upper quartile. ****, $P < 0.0001$.

Using the MPTP-induced macaque model, we seek to avoid postmortem effects of human tissues, thus detecting more faithful transcriptomic changes in PD^{1,2}. MPTP toxicity-induced molecular alterations detected in our study are detailed in the sections “*Differential vulnerability and resilience of DaNs*” and “*Transcriptomic signature of neuronal resilience between DaNs and Gluts*” of the manuscript.

In our introduction part, we did not claim to assay new cell states earlier in disease; instead, consistent with our wording in the introduction part “*post-mortem brains usually reflect the end-stage of PD when DaNs tend to be undersampled in the human specimens*”, we obtained 0.56% of dopaminergic neurons, a proportion higher than

	most human snRNA-seq studies with an unbiased dissection protocol. Furthermore, we leverage this amenable macaque model to control for various factors of macaque individuals (environment, age, sex, etc.) that may confound the interpretability of case-control comparisons, which is usually not strictly controlled with human brain specimens. Our results demonstrate that the vulnerable or resilient dopaminergic neuron types in macaques are in general aligned with those in humans, verifying the usefulness of our model. To avoid confusion, we have now expanded text (lines 250-252) in the section “Cross-species comparison reveals a conserved neuronal resistance gradient”, and added a sentence in the Discussion section (lines 351-353) to explicitly state that our MPTP-treated macaque model 10 months after the MPTP treatment does not detect new states of nigral dopaminergic neurons.
3. A major missing element in this work is a lack of behavioral or pathological characterization of the MPTP-treated monkeys. How did the authors assess if a single dose MPTP injection can produce stable parkinsonian symptoms? A Kurlan score is briefly mentioned in the methods, but no data are presented. The details must be mentioned and data shown. Stable maintenance of PD symptoms is necessary for these types of studies. Some monkeys may show recovery by compensatory mechanisms (e.g. serotonin pathway). Previous studies suggested some guidelines to assess the stability of symptoms for 8 weeks after the MPTP injection. It is also essential to show	We thank the reviewer for this useful comment, and agree that behavioral and pathological characterization is needed for the MPTP-treated macaques. Therefore, we have performed extra analyses, which add confidence that our findings represent generalizable features of parkinsonism, detailed as follows:  1) Macaques treated with MPTP manifested PD-like symptoms including bradykinesia, rigidity and tremor (data not shown). Though there exist several methods to induce parkinsonism in monkeys, in this study we have chosen the unilateral intracarotid injection of MPTP, due to two major advantages of a single injection approach. Firstly, this approach mainly causes an ipsilateral loss of nigrostriatal DaNs whilst avoiding the monkey from paralyzation, easing the post-injection care and increasing the monkey survival rate by reducing the risk of undesired outcomes related to inordinate MPTP exposure³. Secondly, single unilateral intracarotid infusion of MPTP is able to obtain long-term and stable parkinsonian monkeys, as described by previous studies.⁴⁻⁹ 2) MPTP-treated macaques are able to produce stable parkinsonism. After a single dose injection, the Kurlan score peaked at day 20, and became stable throughout the remaining days before macaques were sacrificed (Figure R1). We have now added a Supplementary Table 8 recording the Kurlan scores, as well as a Supplementary

basic pathological analyses of the analyzed tissues. Is there obvious gliosis in SN or PT? Can a rough estimate of the DA loss be obtained? Are there any other brain regions that are noticeably affected?

Figure 1 showing the temporal pattern of Kurlan scores.

Figure R1. Loss of dopaminergic neurons in the macaque model of parkinsonism. **a** Temporal pattern of the Kurlan scores after MPTP injection. Shown are patterns for two MPTP-treated macaques sampled across 17 time points. Kurlan scores after day 210 became stable and were not recorded. **b** IHC images showing TH stainings in the substantia nigra of control and MPTP-treated macaques. Scale bar, 500 μ m. Box plot to the right displays the numbers of TH-positive cells per slice between control and parkinsonian macaques, with the box frames extended from the lower quartile, median, to upper quartile. ****, $P < 0.0001$.

3) Pathological analyses confirm the neuron loss and increased gliosis in the parkinsonian macaques. Immunohistological examination of tyrosine hydroxylase (TH) showed a significant loss of dopaminergic neurons in the MPTP-treated macaques as compared to the control subjects (Figure R1). We estimated the loss of DaNs by counting four slices of MPTP-treated and control subjects separately. As shown in Figure R1, more than 80% of DaNs (an average of 83%) were lost after MPTP administration. Similarly, stainings of GFAP, OLIGO2, and IBA1 demonstrated more notable gliosis in parkinsonian macaques in comparison to controls. All these immunostainings were further consistent with the downstream scRNA-seq analyses which revealed loss of dopaminergic neurons and gain of gliosis in MPTP-treated macaques. We have now added these images as the Supplementary

Figures 1 and 2 in the manuscript.

	Figure R2. Increased gliosis in the macaque model of parkinsonism. IHC images showing stainings of microglia marker IBA1 (a), astrocyte marker GFAP (b), and oligodendrocyte marker OLIGO2 (c) in control (left) and MPTP-treated (right) macaques. Solid and dotted rectangles mark the areas corresponding to the second and third rows of the images, respectively. Bar plots to the right displays the numbers of IBA1-, GFAP-, OLIGO2-positive cells per mm³ between control and parkinsonian macaques. Data are presented as mean ± SD. Scale bars: 500 μm (low magnification), 50 μm (high magnification). ****, $P < 0.0001$; *, $P < 0.05$. 4) Besides substantia nigra and putamen, previous studies reported several other brain regions that were affected by MPTP-induced toxin. For example, the serotonergic neuronal systems were significantly affected by MPTP¹⁰. Moreover, in the MPTP-treated macaque model, the cortical and subcortical regions also displayed a reduction of serotonin transporter activity measured by multiparametric PET¹⁰. Another study quantifying the concentrations of monoamine dopamine, noradrenaline and serotonin in MPTP-treated monkeys found that in addition to the nigrostriatal system, many extrastriatal regions of the subcortex and brainstem also bore a significant loss of noradrenaline¹¹. In the meantime, all three monoamine neuron systems in the cerebral cortex suffered widespread neuronal loss. During the revision, all above-mentioned analyses/discussions have now been incorporated in the manuscript.
4. The authors repeatedly refer to the macaques that received MPTP as “PD.” This is misleading terminology. Injecting a monkey with a toxin is not the same as Parkinson’s disease, which is a human disease. The text and figures should be updated to properly label these monkeys.	We thank the reviewer for pointing out this mistake. We have now changed ‘PD monkey’ into ‘MPTP-treated monkey’, and discriminated between Parkinson’s disease and MPTP-induced neurotoxicity throughout the manuscript (line 20, 21, 49, 66, 75, 107, 112, 117, 126, etc.).
5. One sample (control FM-1128) went	In the manuscript, we have sequenced both cells and nuclei in our data and used them

through single-cell RNA-seq and I assume it was not included in any of the analyses. Can the authors mention the reason, and what differences the scRNA-seq emerged compared with the snRNA-seq data? For example, I see that the proportion of oligodendrocytes is much higher in the snRNA-seq samples compared with scRNA-seq. If it is because of using Myelin Removal Beads II in single-cell suspension preparation compared to single-nucleus suspension preparation, it should be clearly mentioned when describing the proportion of cell-types and its comparison with other studies.	for most analyses such as cell type categorization shown by the global UMAP representation of all cells/nuclei after a batch-correction process (Figure 1). However, analyses relating to cell compositional changes between controls and MPTP-treated macaques were conducted on snRNA-seq samples. This is because nuclei are less prone to damage than cells during isolation and may maintain a better representation of cell type abundance¹². For example, as also mentioned by the reviewer, cell proportions in the scRNA-seq dataset (e.g., decreased number of oligodendrocytes) are more affected by the suspension preparation protocol than those from snRNA-seq. Taken together, we have used both cells and nuclei for the analyses within our manuscript, while focused on the nuclei only when investigating the neuronal and glial proportional changes. All these have been explicitly stated in the manuscript (lines 67-68, lines 100-101).
6. “At the cell type level, DaNs subtypes in our data demonstrated a good match with the molecular taxonomy of DaNs from the human PD patients and controls”. “The two vulnerable types in macaques, SOX6+_CUX2 and SOX6+_RYR3, were matched as whole to the vulnerable subtypes of SOX6_AGTR1 and SOX6_PART1 in humans.” The above-mentioned sentences are ambiguous. How did the authors match human data with macaque data? If they used label transfer, there need to be confusion matrices and/or river plots, along with joint marker	We thank the reviewer for this comment. In fact, we did perform the label transfer analysis for dopaminergic neurons between humans and macaques using SCCAF¹³, as shown in the river plot in Figure 4B. For the sentences mentioned by the reviewer, we are sorry for missing figure references, and have now referred to this figure panel correctly (Figure 4B). In addition to the label transfer analysis, in the revised manuscript we have also provided a joint marker plot (new Supplementary Figure 12; Figure R3) to further corroborate the correspondence of the dopaminergic neurons between humans and macaques. We have now added this new supplemental figure in the manuscript.

plots, to quantitatively demonstrate the quality of the alignment.

Figure R3. Expression of marker genes in human and macaque DaNs. **a** Dot plots showing expression of selected marker genes in macaque (top) and human (bottom) DaNs. Color of the dot represents normalized gene expression, and size represents the percentage of cells expressing a given gene. **b** Sankey diagram displaying the correspondence between DaNs subtypes in our data and those from Kamath et al.

7. How many dopaminergic neurons were detected in cases and controls separately? A total of 730 dopaminergic neurons probably may not be sufficient to recluster them and investigate the transcriptional heterogeneity of dopaminergic neurons. This may be the reason that no PD specific sub-clusters were found. I suggest increasing the number of dopaminergic neurons by including more samples and/or using an enrichment method such as FACS.

We thank the reviewer for pointing out the need for increasing the number of dopaminergic neurons in our study. The 730 dopaminergic neurons from our unbiased dissection protocol consist of 338 controls and 392 cases. Following the reviewer's suggestion, during the revision we have included two additional macaque individuals (a MPTP-treated macaque and a matched control), and enriched for their dopaminergic neurons using fluorescence-activated nucleus sorting (FANS) (new Supplementary Fig. 9; Figure R4). Through this, we have obtained 31,793 sorted nuclei, including 758 dopaminergic neurons, a much higher proportion of DaNs (2.19%±0.58%) than those obtained earlier (0.52%±0.46%) (Supplementary Fig. 9).

We next check whether the seven subtypes identified from the 730 dopaminergic neurons are recapitulated by the 748 FANS-derived dopaminergic neurons. This analysis shows that all the seven subtypes are captured by the sorted neurons, as evidenced by both cell type label transfers and concurrent marker gene expression (new Supplementary Fig. 10; Figure R5). Thus 730 dopaminergic neurons are sufficient for reclustering and subpopulation identification, again highlighting the

quality of our single-cell/nucleus dataset. Moreover, by performing differential abundance analysis (comparison of cell type proportions between controls and diseases), we show a consistent directionality of change for the seven populations in both sc/snRNA-seq and FANS datasets: five enriched and two depleted subtypes in MPTP-treated macaques (Supplementary Fig. 10). Thus, with FANS we validated the cell type diversity and their axis of vulnerability and resilience discovered in the snRNA-seq dataset.

We have now added analyses relating to the FANS-enriched dopaminergic neurons in the revised manuscript, reflected in the text (lines 157-176), Supplementary Figures 9 and 10, and the methods section (lines 428-438).

Figure R4. Nuclei sorted from the substantia nigra of parkinsonian and healthy macaques. **a** Representative FANS plot for enriching midbrain DaNs. The NR4A2 gate was thresholded to select the top 2.3–4.0% of all nuclei. **b** UMAP visualization of all sorted nuclei colored by cell types identified. **c** Stacked bar plots showing the compositions of cell types present across macaque samples. **d** As in **b**, but colored by expression of selected marker genes. **e** Left: stacked bar plot showing the relative contributions of disease conditions to each cell type. Middle: bar plot exhibiting the

number of detected genes in each cell type. Right: stacked bar plot summarizing cell type compositions of the sorted nuclei. **f** Box plot showing proportions of DaNs per replicate for FANS enriched and unbiased samples.

Figure R5. Cellular and molecular diversity of sorted dopaminergic neurons in macaque SN. **a** UMAP visualization of DaNs subclusters colored by cell subtype labels transferred from the snRNA-seq dataset. **b** Left: dot plot showing expression of

	representative marker genes across DaNs subtypes. Color of the dot represents normalized gene expression, and size represents the percentage of cells expressing a given gene. Middle: stacked bar plot exhibiting the proportions of cells in normal (blue) and parkinsonian (red) conditions. Right: violin plot showing cell abundance changes between parkinsonian and control subjects, with the dots representing neighborhoods grouped by cell subtypes. c UMAP visualization of DaNs colored by expression of SOX6 and FOXP2. Violin to the right shows cell abundance changes between two neuronal groups (SOX6⁺ versus FOXP2⁺). d UMAP visualization of DaNs colored by expression of selected genes within the FOXP2 and NR3C1 regulons. e UMAP visualization of DaNs subclusters colored by sample information.
8. In Supplementary Figure 3, the authors showed the proportion of cell-types in cases vs. controls. It is highly recommended to test (e.g. permutation test) whether the proportional alterations are statistically significant or not.	Throughout the manuscript, we used Milo¹⁴, a tool leveraging generalized linear models (GLM) to assess cell neighborhoods and determine differentially abundant cells among conditions, to statistically test the cell type proportional changes in cases versus controls. The results were shown as violin plots with the dots representing neighborhoods grouped by cell types, and can be found in Figure 1g, Figure 2b, Figure 3b, Supplementary Figure 5a and Supplementary Figure 10b.
9. There is some previous evidence regarding the correlation between FOXP2 and neurodegeneration (PMID: 28798667). There is a recently published study on the human prefrontal cortex of PD subjects reporting FOXP2 as an important transcriptomic change in PD (PMID: 36952070). The authors can use the papers when explaining the potential role of FOXP2.	We thank the reviewer for this suggestion. We have now linked our results with the findings from the two papers^{15,16} to further demonstrate the association between FOXP2 and Parkinson's disease. Related text can be found in the Discussion section (lines 342-353).

10. Finding MEIS1 as one of the markers of susceptible excitatory neuron subtypes could be interesting. One possible explanation could be that MEIS1 is one of the known genes in restless legs syndrome (RLS), a disease with disruption in dopamine levels. Several epidemiological studies reported that RLS risk is higher among PD patients and PD is more commonly seen in patients with RLS compared to the general population (PMID: 31004074).	We thank the reviewer for pointing out the biological significance of MEIS1. For glutamatergic excitatory neurons, based on our data, a MEIS1+ subtype and the other MEIS2+ subtype were clearly susceptible to loss in MPTP-treated macaques. These two genes were related to the brain developmental process¹⁷ and striatal medium spiny neuronal apoptosis¹⁸. In our revised manuscript, we have also described the role of MEIS1 in restless legs syndrome and its potential relevance with PD susceptibility (lines 218-219).
11. The authors mention that CALB1 is not a marker of the resilient neurons, but many papers studying DA neurons in primates have seen the CALB1 gradient, including single cell data from PMID 35513515 in macaque. Can the authors explain the discrepancy with their data? Is it due to low numbers of cells, or perhaps some differences in the genome reference or alignment pipelines used in the different studies?	We thank the reviewer for this point. Though CALB1 was shown to be highly expressed in human resilient neurons, its expression is not so prominent in the resilient neurons of macaques, including macaques from our data and from Kamth et al. 2022¹⁹ (PMID 35513515 as mentioned by the reviewer). As shown in the Figure R6 below, in comparison to the human data which has a number of CALB1+ resilient neurons, both macaque datasets are not enriched for CALB1+ neurons. Similar results can also be found in another published dataset profiling macaque DaNs (GSE178265). Similarly, another study comparing the proportions of CALB1+ DaNs between control and MPTP-induced parkinsonian macaques revealed only a slight increase in CALB1+TH+ cells after MPTP treatment, indicating the restricted expression of CALB1 across PD-resistant DaNs in macaques²⁰. Of note, even in the human data, the proportion of CALB1+ neurons is relatively low, as compared to other resilience markers such as FOXP2. We thus believe that the inconsistent expression of CALB1 between humans and macaques are not caused by technical artifacts (common to all macaque studies to our knowledge) and possibly reflect the low number of CALB1+ cells that can be easily undersampled. As a result, in the present study we focus on FOXP2, which is more widely expressed in the resilient neurons and functionally implicated in neurodegeneration.

Figure R6. UMAP visualizations of dopaminergic neurons from this study, published macaque (GSE178256) and human (GSE178256) studies, as colored by expression of selected genes: *SOX6* (left), *FOXP2* (middle) and *CALB1* (right).

In the revised manuscript, we have now mentioned this point in the Main text (lines 130-131) and discussed it in the Discussion section (lines 373-380).

Minor points

1. “we also found four PD risk genes (SH3GL2, LRRK2, STAB1, BST1), and an Alzheimer’s disease risk gene (CR1), all of which were highly expressed in these vulnerable DaNs subclusters”. Did the authors analyze all PD risk genes in both vulnerable and resistant subtypes? If so, the list of genes should be mentioned in a supplementary table.	We thank the reviewer for this question. We have now added a Supplementary Table 2 to list all PD and AD risk genes tested in our manuscript, which were collated from published studies^{21,22}.
2. I assume that the weight of the monkeys was not considered when determining the dose of MPTP injection? Can the authors justify this in the methods?	We did not consider the weight of the monkeys, and the dose of MPTP injection was constant (3mg) for all monkeys. This strategy was chosen after considering the outcomes from earlier studies. Specifically, due to the high first-pass extraction of MPTP, the major determinant of direct intra-carotid MPTP dosing is the brain volume that does not correlate with the body weight³. An earlier study has demonstrated that use of similar absolute MPTP doses, defined as the total amount of MPTP salt administered, led to similar parkinsonism symptoms but better predicts severe acute neurotoxicity. By contrast, a dose of injection proportional to the body weight may better suit system injections, such as intramuscular or intravenous injection³. To justify this in the manuscript, we have now added this information to the Methods section (lines 394-396).
3. A total number of regulons specific to the dopaminergic neuron subtypes identified in this dataset should be mentioned.	We thank the reviewer for this point. In total, there are 43 regulons found to be associated with vulnerability or resilience of dopaminergic neurons. We have now enumerated these regulons in the Supplementary Table 3 and referred to it in the manuscript (line 146).

4. It is mentioned that the STAB1 gene is a PD-risk gene. Stabilin 1 (STAB1) is not a PD risk gene and apparently, it should be SATB1²¹ (PMID: 31701892).	We thank the reviewer for pointing out the mistake. It was an error of writing. We have now corrected this in the manuscript (line 139).
5. The authors mentioned that “They highly expressed KCNJ6, a marker gene relating to vulnerability in PD²⁴”. However, according to the conclusion of reference 24, GIRK2 (KCNJ6) is unlikely to be a marker to distinguish between dorsal and ventral tier SNpc dopaminergic neurons and is not correlated with differential vulnerability. The authors should either remove it or cite a different paper.	We thank the review for this point. Based on our data, KCNJ6 was enriched in the two SOX6+ vulnerable dopaminergic neuron subtypes (Fig. 2b), as also confirmed by immunostaining of KCNJ6 in the ventral tier of substantia nigra (Supplementary Fig. 11a). For reference 24, its conclusion is indeed consistent with our result, that is, GIRK2 (KCNJ6) is potentially correlated with the differential vulnerability of subgroups of midbrain DaNs. We also found another paper which clearly stated that the vulnerable DaNs in the SN have a particularly pronounced D2-AR/GIRK2 feedback regulation in comparison with more resistant VTA DaNs²³. All these indicate that KCNJ6 may serve as a potential marker for DaNs vulnerability. We have now changed the citation in our study (lines 134-135).
6. The authors should be more consistent in using terms. For example, 10x (10X), Cell Ranger (Cellranger), etc.	We thank the reviewer for pointing out these. We have now examined the various technical terms used in our manuscript and corrected them accordingly.
7. A full list of antibodies and their characteristics must be included as Supplementary Material.	We thank the reviewer for pointing out the missing details of antibodies. We have now listed the antibodies and described their details in our study (lines 524-529).

8. Typographic:  - Line 102 “Board” - Line 121 “in partial” - Line 315 “as as” - Line 413 “ell population” 	We thank the reviewer for this point. We have now carefully checked all text and corrected typos throughout the manuscript (including those mentioned by the reviewer).
Reviewer #2	
In this paper, the authors used single-cell nucleus transcriptomic profiling technology to assess the molecular heterogeneity of midbrain dopaminergic neurons and striatal projection neurons in the putamen of rhesus macaques. To determine if specific genes expression is altered in parkinsonian state, the ventral midbrain and putamen of both control and acutely MPTP-treated parkinsonian monkeys were examined, with the ultimate goal of determining if the expression changes of specific transcriptomes confer neuroprotection or susceptibility of various subtypes of dopamine and striatal neurons to MPTP neurotoxicity. Although similar approaches were used to associate molecular heterogeneity and sensitivity of midbrain DAergic neurons to PD-induced degeneration in humans and	We thank the reviewer for the accurate summary and positive evaluation of our work. Please find our point-by-point response below, together with extra analyses, new figures, tables, and revised text.

mice, the present study complements the published literature with the addition of data from rhesus monkeys. Regarding the striatal data, the present results are in part confirmatory to those recently published by He et al (Current Biology, 2021; 31:5473), but the information about striatal changes in gene expression in the putamen of MPTP-treated monkeys is new. Overall, this is a timely study that uses cutting-edge transcriptomic approaches. Despite the fact that a significant amount of data is confirmatory of the published literature, part of the content of this paper extends these previous findings and further strengthens that the molecular heterogeneity of midbrain DA neurons may contribute to their selective resistance or sensitivity to MPTP-induced neurotoxicity. However, there are significant limitations that must be addressed:

1. The validity of the animal model used to assess pathogenesis of midbrain dopamine neurons. Although MPTP-induced toxicity of dopamine cells follows a pattern reminiscent of what has been observed in human PD, the acute nature of the intracarotid MPTP monkey model used in this study has some limitations.

We thank the reviewer for this comment. Despite the acute nature of MPTP-induced toxicity, our MPTP-treated macaque model produced stable parkinsonian symptoms along 10 months from MPTP injection. Specifically, though lacking Lewy bodies is a classic opposition to the MPTP-induced model, several studies have shown that the mRNA and protein of α -synuclein, the main component of Lewy bodies, are upregulated in SNc after the MPTP treatment. One week after a single injection of MPTP in squirrel monkeys, α -synuclein mRNA and protein were markedly enhanced in neuronal fibers instead of nigral cell bodies²⁴. One month after MPTP injection, α -

A chronic low dose MPTP-treated monkey model would have been more valuable. Another major concern about the MPTP monkey model is the fact that MPTP-treated monkeys do not exhibit Lewy bodies which is an important shortcoming of this when this model (or any other neurotoxin-based model of PD) is used to assess dopamine cells pathogenesis. These concerns limit the significance of the present results as they relate to the human PD condition.

synuclein mRNA and protein remained elevated and are detectable in a significant number of cell bodies. Furthermore, approximately 80% of the dopaminergic cell bodies that survived MPTP toxicity were stained positive for α -synuclein²⁵.

In our study, we also showed that though being absent of Lewy bodies, our macaque model of parkinsonism mirrored a number of human PD features at the behavioral, pathological, cellular and molecular levels. Details are described as follows:

1) Macaques treated with MPTP manifested PD-like symptoms including bradykinesia, rigidity and tremor (data not shown). Though there exist several methods to induce parkinsonism in monkeys, in this study we have chosen the unilateral intracarotid injection of MPTP, due to two major advantages of a single injection approach. Firstly, this approach mainly causes an ipsilateral loss of nigrostriatal DaNs whilst avoiding the monkey from paralyzation, easing the post-injection care and increasing the monkey survival rate by reducing the risk of undesired outcomes related to inordinate MPTP exposure³. Secondly, single unilateral intracarotid infusion of MPTP is able to obtain long-term and stable parkinsonian monkeys, as described by previous studies.⁴⁻⁹

2) MPTP-treated macaques are able to produce stable parkinsonism. After a single dose injection, the Kurlan score peaked at day 20, and became stable throughout the remaining days before macaques were sacrificed (Figure R1). We have now added a Supplementary Table 8 recording the Kurlan scores, as well as a Supplementary Figure 1 showing the temporal pattern of Kurlan scores.

Figure R1. Loss of dopaminergic neurons in the macaque model of parkinsonism. **a** Temporal pattern of the Kurlan scores after MPTP injection. Shown are patterns for two MPTP-treated macaques sampled across 17 time points. Kurlan scores after day 210 became stable and were not recorded. **b** IHC images showing TH stainings in the substantia nigra of control and MPTP-treated macaques. Scale bar, 500 μ m. Box plot to the right displays the numbers of TH-positive cells per slice between control and parkinsonian macaques, with the box frames extended from the lower quartile, median, to upper quartile. ****, $P < 0.0001$.

3) Pathological analyses confirm the neuron loss and increased gliosis in the parkinsonian macaques. Immunohistological examination of tyrosine hydroxylase (TH) showed a significant loss of dopaminergic neurons in the MPTP-treated macaques as compared to the control subjects (Figure R1). We estimated the loss of DaNs by counting four slices of MPTP-treated and control subjects separately. As shown in Figure R1, more than 80% of DaNs (an average of 83%) were lost after MPTP administration. Similarly, stainings of GFAP, OLIGO2, and IBA1 demonstrated more notable gliosis in parkinsonian macaques in comparison to controls. All these immunostainings were further consistent with the downstream scRNA-seq analyses which revealed loss of dopaminergic neurons and gain of gliosis in MPTP-treated macaques. We have now added these images as the Supplementary Figures 1 and 2 in the manuscript.

Figure R2. Increased gliosis in the macaque model of parkinsonism. IHC images showing stainings of microglia marker IBA1 (a), astrocyte marker GFAP (b), and

oligodendrocyte marker OLIGO2 (c) in control (left) and MPTP-treated (right) macaques. Solid and dotted rectangles mark the areas corresponding to the second and third rows of the images, respectively. Bar plots to the right displays the numbers of IBA1-, GFAP-, OLIGO2-positive cells per mm³ between control and parkinsonian macaques. Data are presented as mean ± SD. Scale bars: 500 μm (low magnification), 50 μm (high magnification). ****, $P < 0.0001$; *, $P < 0.05$.

4) Besides substantia nigra and putamen, previous studies reported several other brain regions that were affected by MPTP-induced toxin. For example, the serotonergic neuronal systems were significantly affected by MPTP¹⁰. Moreover, in the MPTP-treated macaque model, the cortical and subcortical regions also displayed a reduction of serotonin transporter activity measured by multiparametric PET¹⁰. Another study quantifying the concentrations of monoamine dopamine, noradrenaline and serotonin in MPTP-treated monkeys found that in addition to the nigrostriatal system, many extrastriatal regions of the subcortex and brainstem also bore a significant loss of noradrenaline¹¹. In the meantime, all three monoamine neuron systems in the cerebral cortex suffered widespread neuronal loss.

5) Consistent with the above behavioral and pathological analyses, our single-cell analyses revealed the association between MPTP-induced changes and neuronal vulnerability and resilience, including the core susceptibility axis and regulatory units shared between PD patients and MPTP-induced macaque models. This again showcases the validity of our animal model in recapitulating a number of important cell type characteristics found in human subjects.

During the revision, all above-mentioned analyses/discussions have now been incorporated in the manuscript.

2. In the ventral midbrain, the authors compare transcriptomic data collected from different types of DA cells and Excitatory neurons. It is not clear based on the information provided what population of neurons account for these excitatory cells. In the SNc and VTA, subsets of neurons co-express DA and vGluT2. Are these the neurons the authors refer to or is it a population of non-dopaminergic glutamatergic neurons. Please provide more information about the transmitter co-expression phenotype of this neuronal subpopulation.	We thank the reviewer for this suggestion. Based on our data, there are three broad neuron types in the substantia nigra: dopaminergic neurons (DaNs) (7 clusters), glutamatergic excitatory neurons (6 clusters), and inhibitory neurons (5 clusters). We would like to note that although some DaNs co-express DA and vGluT2, we still classified them as dopaminergic, considering that their molecular traits, as a whole, resembled other typical dopaminergic neurons as determined by unsupervised clustering (Figure 1c, e and Supplementary Fig. 3d, e). In the section “Transcriptomic signature of neuronal resilience between DaNs and Gluts”, we focused on the neurons which do not belong to dopaminergic neurons and highly expressed SLC17A6. To disambiguate these terms, we have now modified our wording for these neuron types as “dopaminergic neurons”, “glutamatergic excitatory neurons”, and “inhibitory neurons”.
3. Although of potential interest the striatal data are in large part confirmatory of recent findings published by He et al in Current Biology. The authors must cite and discuss the content of this paper in detail.	We thank the reviewer for pointing out this paper. In our revised manuscript, we have now cited He et al. 2021, and discussed the consistency between our results and those from that paper (lines 91-92, lines 271-272).

4. In lines 164-165, the authors state: “The vulnerable DaNs subtypes upregulated an extensively overlapped set of genes, indicating a similar response induced by PD pathology”. This statement is confusing because the so-called vulnerable DaNs in the SN of MPTP treated monkeys have died and are not part of the tissue analyzed from these monkeys. If some neurons remain in the SN of these parkinsonian animals, shouldn’t they be categorized as resilient (instead of vulnerable) to MPTP neurotoxicity? This is a limitation of the animal model used in the study.	Here we defined vulnerable neurons as those enriched in healthy macaques but depleted in MPTP-treated macaques. In other words, we speculate these neurons to undergo a significant loss, as assessed by a statistical test of cell neighborhoods using Milo¹⁴ by comparing cell abundance between controls and MPTP-induced macaques. This definition of vulnerable neurons can be also found in previous studies for example: Single-cell genomic profiling of human dopamine neurons identifies a population that selectively degenerates in Parkinson’s disease (Kamath et al. 2022)
Minor points	
1. Avoid using the terminology “PD macaques” when referring to the MPTP monkey model because these animals do not have PD. For this reason, the MPTP monkey model is often considered as a model of parkinsonism instead of PD.	We thank the reviewer for pointing out this mistake. We have now changed ‘PD macaques’ into ‘MPTP-treated macaques’ or “model of parkinsonism”, and discriminated between Parkinson’s disease and MPTP-induced neurotoxicity throughout the manuscript (line 20, 21, 49, 66, 75, 107, 112, 117, 126, etc.).

2. Lines 15-16: Wording of some statements must be improved. For instance, in the abstract the authors say “.....progressive loss of neurons in the nigrostriatal system including the substantia nigra (SN) and putamen (PT)”. This statements sounds as if both the SN and PT contains nigrostriatal neurons	We thank the reviewer for this point. We have now rewritten the sentence as “The degenerative process in Parkinson’s disease (PD) causes a progressive loss of dopaminergic neurons (DaNs) in the nigrostriatal system” to make it clear that the loss of neurons mainly occurs for dopaminergic neurons (lines 16-17).
3. Line 20: “Our PD model recapitulates the pathologic features in nature PD”. This overstatement is incorrect. The MPTP model recapitulates “some” of the pathologic features in nature PD.	We thank the reviewer for pointing out this overstatement. We have now changed this sentence to “Our primate model of parkinsonism recapitulates important pathologic features in nature PD” (lines 21-22).
4. Line 24: “excitatory neurons” If these are DA/vGluT2 neurons, these cells should be named accordingly to avoid confusion. As currently presented, they are considered as a subset of non-DA glutamatergic neurons in the SN. Is it the case?	We thank the reviewer for this point. There are some DA/vGluT2 neurons and we classify them as dopaminergic, considering that their molecular traits, as a whole, resembled other typical dopaminergic neurons as determined by unsupervised clustering (Figure 1c, e and Supplementary Fig. 3d, e). In the section “Transcriptomic signature of neuronal resilience between DaNs and Gluts”, we focused on the neurons which do not belong to dopaminergic neurons and highly expressed SLC17A6. To disambiguate these terms, we have now modified our wording for these neuron types as “dopaminergic neurons”, “glutamatergic excitatory neurons”, and “inhibitory neurons”.
Reviewer #3	
This is a study of the transcriptomic profile of nigral and putamenal cells in non-human primates using the MPTP model of	We thank the reviewer for summarizing our manuscript and valuing our data and findings. Please find our point-by-point response below, together with extra analyses, new figures, tables, and revised text.

Parkinson's disease. To determine transcript changes, particularly underlying cell vulnerability, 10X genomics chromium platform for sc and sn-RNAseq was applied comparing samples from parkinsonian and control macaques. Data were analyzed with established methodologies. Most significant results are that SOX6+ expression was associated with DA neuron vulnerability while SOX6- cells especially those expressing FOXP2 and SORCS3 corresponded with higher resistance. The available human data (SOX6-CALB1) support these results. The topographic distribution of the profiled vulnerable and resilient neurons was correlated with the typical distribution of PD pathology between SNc ventral and dorsal tiers. Analyses of the regulated pathways also showed interesting transcript regulation, including support for the presence of inflammation. The study also identified other genes with different expression patterns in vulnerable versus resistant cells.

As expected from this type of descriptive study, comparisons across species and cell types are important. Results showed some misalignments in comparisons with human data, for example in microglia, but there were

large similarities in the nigral DA neurons. Overall, these data can contribute to characterizing mechanisms of cell death and survival in PD. Some concerns come from the following points:	
1. The activation of immune responses (microglia and astrocytes) in SNc, and the striatum as well, are more difficult to interpret in relation to PD considering that samples derived from an acute toxin model.	We thank the reviewer for this point. Although activation signatures can be possibly induced by the acute toxin model, most of the pathways we identified in the manuscript were not likely caused by neurotoxin. For example, a number of genes that were upregulated in astrocytes and microglia of human PD studies were also recapitulated by our macaque model of parkinsonism, indicating a concurrently perturbed signature in MPTP-treated macaques and human PD patients (Figure R7). One example is from the human microglia where GPNMB and LRRK2 (two PD risk genes upregulated in MPTP-treated macaques) are shown to be robustly overrepresented in human PD. This is also true for astrocytes where a number of activation signatures are common to both our model of parkinsonism and human PD patients. We also observed some degree of inconsistencies in direction of gene expression changes, possibly due to MPTP treatment-specific responses or inherent cross-species differences. Therefore, though there may exist immune response-related activation caused by MPTP toxicity, our primate model of parkinsonism still captures most of the genuine signatures that were also previously discovered in human PD patients.

Figure R7: Comparison of activated genes in glia of MPTP-treated macaques and human PD patients. **a** Scatter plot showing the average expression of immune response- and PD risk-related genes in microglia of control and MPTP-treated macaques (left) or human PD patients (middle and right). Colored dots represent significantly up-regulated genes: red, immune response genes; blue, PD risk genes. Text highlights co-activated genes in both human PD patients and MPTP-treated macaques. **b** As in **a**, but for astrocytes.

2. There are many transcriptomic changes described in the putamen, but this section is not well organized. Genes related to synaptic

We thank the reviewer for pointing out this. We have now reorganized the part relating to transcriptomic changes in the putamen. Specifically, we grouped the molecular alterations induced in the putamen into two categories: up-/down-regulated genes caused by parkinsonism, and cell type-shared/-specific genes between dSPN and

transmission, neurodegeneration, inflammation, etc. are intermingled making it difficult to extract the most relevant information. In addition, some data are difficult to put in the context of PD, such as the regulation in striatal neurons of genes related to apoptosis.	iSPN. In each category, we have now restricted discussion of expression programs and enriched GO terms to synapses, protein ubiquitination, projection organization, and other pathways that are of potential relevance to regulation in Parkinson's disease. We also removed some ambiguous links to striatal neurons and Parkinson's disease. The revised text can be found in the revised manuscript (lines 290-292, lines 295-297, lines 312-314).
3. The discussion suffers of miscalibration of data generated in the primate MPTP model of PD. Although the model can be said that is, quote, ‘representative of PD pathophysiology’, clearly it is not representative of PD pathogenesis, which is most relevant to a transcriptomic analysis. One recognizes that the vulnerability of certain neurons, as an inherent condition, may be determined by some shared genes between human and non-human primates independent of the insult. But having a different causative mechanism plays a major role in the determinants of cell vulnerability to degeneration and death, and likely because of different gene regulation. At a minimum, studies to determine if the transcript changes in this animal model are also found in the human disease are necessary for validation of these data.	We thank the reviewer for this comment, and agree that our macaque model, though producing stable parkinsonian symptoms (e.g., stable Kurlan scores), is not “representative of PD pathophysiology in humans”. We now rephrase the wording of our macaque “PD model” into ‘MPTP-treated macaques’ or “model of parkinsonism”. Nevertheless, this MPTP-induced model mirrored a number of human PD features at the behavioral, pathological, cellular and molecular levels, described as follows:  1) Macaques treated with MPTP manifested PD-like symptoms including bradykinesia, rigidity and tremor (data not shown). Though there exist several methods to induce parkinsonism in monkeys, in this study we have chosen the unilateral intracarotid injection of MPTP, due to two major advantages of a single injection approach. Firstly, this approach mainly causes an ipsilateral loss of nigrostriatal DaNs whilst avoiding the monkey from paralyzation, easing the post-injection care and increasing the monkey survival rate by reducing the risk of undesired outcomes related to inordinate MPTP exposure³. Secondly, single unilateral intracarotid infusion of MPTP is able to obtain long-term and stable parkinsonian monkeys, as described by previous studies.⁴⁻⁹ 2) MPTP-treated macaques are able to produce stable parkinsonism. After a single dose injection, the Kurlan score peaked at day 20, and became stable throughout the remaining days before macaques were sacrificed (Figure R1). We have now added a Supplementary Table 8 recording the Kurlan scores, as well as a Supplementary Figure 1 showing the temporal pattern of Kurlan scores.

Figure R1. Loss of dopaminergic neurons in the macaque model of parkinsonism. **a** Temporal pattern of the Kurlan scores after MPTP injection. Shown are patterns for two MPTP-treated macaques sampled across 17 time points. Kurlan scores after day 210 became stable and were not recorded. **b** IHC images showing TH stainings in the substantia nigra of control and MPTP-treated macaques. Scale bar, 500 μ m. Box plot to the right displays the numbers of TH-positive cells per slice between control and parkinsonian macaques, with the box frames extended from the lower quartile, median, to upper quartile. ****, $P < 0.0001$.

3) Pathological analyses confirm the neuron loss and increased gliosis in the parkinsonian macaques. Immunohistological examination of tyrosine hydroxylase (TH) showed a significant loss of dopaminergic neurons in the MPTP-treated macaques as compared to the control subjects (Figure R1). We estimated the loss of DaNs by counting four slices of MPTP-treated and control subjects separately. As shown in Figure R1, more than 80% of DaNs (an average of 83%) were lost after MPTP administration. Similarly, stainings of GFAP, OLIGO2, and IBA1 demonstrated more notable gliosis in parkinsonian macaques in comparison to controls. All these immunostainings were further consistent with the downstream scRNA-seq analyses which revealed loss of dopaminergic neurons and gain of gliosis in MPTP-treated macaques. We have now added these images as the Supplementary Figures 1 and 2 in the manuscript.

Figure R2. Increased gliosis in the macaque model of parkinsonism. IHC images

	showing stainings of microglia marker IBA1 (a), astrocyte marker GFAP (b), and oligodendrocyte marker OLIGO2 (c) in control (left) and MPTP-treated (right) macaques. Solid and dotted rectangles mark the areas corresponding to the second and third rows of the images, respectively. Bar plots to the right displays the numbers of IBA1-, GFAP-, OLIGO2-positive cells per mm³ between control and parkinsonian macaques. Data are presented as mean ± SD. Scale bars: 500 μm (low magnification), 50 μm (high magnification). ****, $P < 0.0001$; *, $P < 0.05$. 4) Besides substantia nigra and putamen, previous studies reported several other brain regions that were affected by MPTP-induced toxin. For example, the serotonergic neuronal systems were significantly affected by MPTP¹⁰. Moreover, in the MPTP-treated macaque model, the cortical and subcortical regions also displayed a reduction of serotonin transporter activity measured by multiparametric PET¹⁰. Another study quantifying the concentrations of monoamine dopamine, noradrenaline and serotonin in MPTP-treated monkeys found that in addition to the nigrostriatal system, many extrastriatal regions of the subcortex and brainstem also bore a significant loss of noradrenaline¹¹. In the meantime, all three monoamine neuron systems in the cerebral cortex suffered widespread neuronal loss. 5) Both neurons and glia show consistent molecular changes induced by MPTP in macaques or by PD in human patients. For neuronal cells, our single-cell analyses revealed the association between MPTP-induced changes and neuronal vulnerability and resilience in humans, including the core susceptibility axis and regulatory units shared between PD patients and MPTP-induced macaque models (Fig. 4b, c, d). For glial cells, as mentioned in 3.1 (Figure R7), key parkinsonism-causing changes in genes and pathways are shared between macaques and humans. All these showcase the validity of our animal model in recapitulating a number of important cell type characteristics found in human subjects. During the revision, all above-mentioned analyses/discussions have now been incorporated in the manuscript.
4. Comments related to the functional	We thank the reviewer for pointing out this over-discussed topic. We have now

diversity of the putamen, linking dopamine neuron subtype connections to clinical features in patients are not only beyond the scope of this study but also not well founded.

removed the discussion part with respect to functional diversity of the putamen.

1. Zhu, Y., Wang, L., Yin, Y. & Yang, E. Systematic analysis of gene expression patterns associated with postmortem interval in human tissues. *Sci. Rep.* **7**, 5435 (2017).
2. Ferreira, P. G. *et al.* The effects of death and post-mortem cold ischemia on human tissue transcriptomes. *Nat. Commun.* **9**, 490 (2018).
3. Norris, S. A. *et al.* Severe acute neurotoxicity reflects absolute intra-carotid 1-methyl-4-phenyl-1,2,3,6-tetrahydropyridine dose in non-human primates. *J. Neurosci. Methods* **366**, 109406 (2022).
4. Kordower, J. H., Liu, Y. T., Winn, S. & Emerich, D. F. Encapsulated PC12 cell transplants into hemiparkinsonian monkeys: a behavioral, neuroanatomical, and neurochemical analysis. *Cell Transplant.* **4**, 155–171 (1995).
5. Palombo, E. *et al.* Local cerebral glucose utilization in monkeys with hemiparkinsonism induced by intracarotid infusion of the neurotoxin MPTP. *J. Neurosci.* **10**, 860–869 (1990).
6. Schneider, J. S., McLaughlin, W. W. & Roeltgen, D. P. Motor and nonmotor behavioral deficits in monkeys made hemiparkinsonian by intracarotid MPTP infusion. *Neurology* **42**, 1565–1572 (1992).
7. Clarke, C. E., Boyce, S., Robertson, R. G., Sambrook, M. A. & Crossman, A. R. Drug-induced dyskinesia in primates rendered hemiparkinsonian by intracarotid administration of 1-methyl-4-phenyl-1,2,3,6-tetrahydropyridine (MPTP). *J. Neurol. Sci.* **90**, 307–314 (1989).
8. Emborg-Knott, M. E. & Domino, E. F. MPTP-Induced hemiparkinsonism in nonhuman primates 6-8 years after a single unilateral intracarotid dose. *Exp. Neurol.* **152**, 214–220 (1998).

9. Benazzouz, A., Gross, C., Féger, J., Boraud, T. & Bioulac, B. Reversal of rigidity and improvement in motor performance by subthalamic high-frequency stimulation in MPTP-treated monkeys. *Eur. J. Neurosci.* **5**, 382–389 (1993).
10. Kanazawa, M., Ohba, H., Nishiyama, S., Kakiuchi, T. & Tsukada, H. Effect of MPTP on Serotonergic Neuronal Systems and Mitochondrial Complex I Activity in the Living Brain: A PET Study on Conscious Rhesus Monkeys. *J. Nucl. Med.* **58**, 1111–1116 (2017).
11. Pifl, C., Schingnitz, G. & Hornykiewicz, O. Effect of 1-methyl-4-phenyl-1,2,3,6-tetrahydropyridine on the regional distribution of brain monoamines in the rhesus monkey. *Neuroscience* **44**, 591–605 (1991).
12. Bakken, T. E. *et al.* Single-nucleus and single-cell transcriptomes compared in matched cortical cell types. *PLoS One* **13**, e0209648 (2018).
13. Miao, Z. *et al.* Putative cell type discovery from single-cell gene expression data. *Nat. Methods* **17**, 621–628 (2020).
14. Dann, E., Henderson, N. C., Teichmann, S. A., Morgan, M. D. & Marioni, J. C. Differential abundance testing on single-cell data using k-nearest neighbor graphs. *Nat. Biotechnol.* **40**, 245–253 (2022).
15. Lin, L.-C., Cole, R. C., Greenlee, J. D. W. & Narayanan, N. S. A Pilot Study of Ex Vivo Human Prefrontal RNA Transcriptomics in Parkinson's Disease. *Cell. Mol. Neurobiol.* **43**, 3037–3046 (2023).
16. Oswald, F. *et al.* The FOXP2-Driven Network in Developmental Disorders and Neurodegeneration. *Front. Cell. Neurosci.* **11**, 212 (2017).
17. Owa, T. *et al.* Meis1 Coordinates Cerebellar Granule Cell Development by Regulating Pax6 Transcription, BMP Signaling and Atoh1 Degradation. *J. Neurosci.* **38**, 1277–1294 (2018).
18. Yang, L. *et al.* Transcriptional profiling reveals the transcription factor networks regulating the survival of striatal neurons. *Cell Death Dis.* **12**, 262 (2021).
19. Kamath, T. *et al.* Single-cell genomic profiling of human dopamine neurons identifies a population that selectively degenerates in Parkinson's

- disease. *Nat. Neurosci.* **25**, 588–595 (2022).
20. Dopeso-Reyes, I. G. *et al.* Calbindin content and differential vulnerability of midbrain efferent dopaminergic neurons in macaques. *Front. Neuroanat.* **8**, 146 (2014).
 21. Nalls, M. A. *et al.* Identification of novel risk loci, causal insights, and heritable risk for Parkinson’s disease: a meta-analysis of genome-wide association studies. *Lancet Neurol.* **18**, 1091–1102 (2019).
 22. Jansen, I. E. *et al.* Genome-wide meta-analysis identifies new loci and functional pathways influencing Alzheimer’s disease risk. *Nat. Genet.* **51**, 404–413 (2019).
 23. Duda, J., Pötschke, C. & Liss, B. Converging roles of ion channels, calcium, metabolic stress, and activity pattern of Substantia nigra dopaminergic neurons in health and Parkinson’s disease. *J. Neurochem.* **139 Suppl 1**, 156–178 (2016).
 24. Purisai, M. G., McCormack, A. L., Langston, W. J., Johnston, L. C. & Di Monte, D. A. Alpha-synuclein expression in the substantia nigra of MPTP-lesioned non-human primates. *Neurobiol. Dis.* **20**, 898–906 (2005).
 25. Porras, G., Li, Q. & Bezard, E. Modeling Parkinson’s disease in primates: The MPTP model. *Cold Spring Harb. Perspect. Med.* **2**, a009308 (2012).

REVIEWERS' COMMENTS

Reviewer #1 (Remarks to the Author):

The authors have addressed all the major points. The inclusion of the behavioral assessments of MPTP effectiveness are particularly helpful to bolstering the significance and confidence in the findings.

Reviewer #2 (Remarks to the Author):

The authors have satisfactorily addressed my concerns. I do not have more comments to add.

Reviewer #3 (Remarks to the Author):

The authors' revision improved the quality of the paper. I do not have additional comments.

Stella M. Papa